# Adaptive Partial Train Speed Trajectory Optimization

**Zhaoxiang Tan [1], Shaofeng Lu [1,*] , Kai Bao [1], Shaoning Zhang [1,2] , Chaoxian Wu [1,2] , Jie Yang [3] and Fei Xue [1]**

[1] Department of Electrical and Electronic Engineering, Xi'an Jiaotong-Liverpool University, Suzhou 215123, China; zhaoxiangtan@foxmail.com (Z.T.); kaibao5230@gmail.com (K.B.); shaoningzhang716@gmail.com (S.Z.); Chaoxian.Wu@Liverpool.ac.uk (C.W.); Fei.Xue@xjtlu.edu.cn (F.X.)
[2] Department of Electrical Engineering and Electronics, The University of Liverpool, Liverpool L69 3BX, UK
[3] School of Electrical Engineering and Automation, Jiangxi University of Science and Technology, Ganzhou 341000, China; yangjie2013@bjtu.edu.cn
* Correspondence: Shaofeng.Lu@xjtlu.edu.cn; Tel.: +86-512-8816-7110

**Abstract:** Train speed trajectory optimization has been proposed as an efficient and feasible method for energy-efficient train operation without many further requirements to upgrade the current railway system. This paper focuses on an adaptive partial train speed trajectory optimization problem between two arbitrary speed points with a given traveling time and distance, in comparison with full speed trajectory with zero initial and end speeds between two stations. This optimization problem is of interest in dynamic applications where scenarios keep changing due to signaling and multi-train interactions. We present a detailed optimality analysis based on Pontryagin's maximum principle (PMP) which is later used to design the optimization methods. We propose two optimization methods, one based on the PMP and another based on mixed-integer linear programming (MILP), to solve the problem. Both methods are designed using heuristics obtained from the developed optimality analysis based on the PMP. We develop an intuitive numerical algorithm to achieve the optimal speed trajectory in four typical case scenarios; meanwhile, we propose a new distance-based MILP approach to optimize the partial speed trajectory in the same scenarios with high modeling precision and computation efficiency. The MILP method is later used in a real engineering speed trajectory optimization to demonstrate its high computational efficiency, robustness, and adaptivity. This paper concludes with a comparison of both methods in addition to the widely applied pseudospectral method and propose the future work of this paper.

**Keywords:** Pontryagin's Maximum Principle (PMP); speed trajectory optimization; Mixed-Integer Linear Programming (MILP); energy-efficient train operation; motor efficiency; Pseudospectral method

## 1. Introduction

With the increasing energy demand and more emphasis put on the carbon footprint, energy-saving is becoming a hot spot in rail transportation. In order to obtain lower energy consumptions, many researchers have focused on energy-efficiency enhancement technologies in both academia and industry. In these studies, such as the earlier works by [1,2], how to use the traction energy in a more efficient way and to use electrical braking to recycle the kinetic energy are studied. With few requirements to improve the existing infrastructures, the speed trajectory optimization is regarded as an effective method to reduce the energy consumption in rail systems. In this paper, the regenerative braking is utilized to reduce the total net energy consumption. In particular, this paper explicitly considers the motor efficiency during both traction and braking procedures.

In previous studies, [3] demonstrated that the constant efficiency of motor plays a key role in the partial speed trajectory optimization problem. In most cases, a constant efficiency does not fully

reflect the efficiency of electric motors in real operation, and taking more motor characteristics of motor may enable the model to generate more precise results, reflecting the actual energy consumption in short-term dynamic operations. The current constant-efficiency assumption is based on a general observation that the motor efficiency can be considered constant for the long-term operations of typical railway vehicles. Based on our study outcomes, we wish to offer an optimal train control strategy for the long-term operations, which has been adopted by most existing studies. We considered the current proposed methods can be applied to most traditional tractions where the motor efficiency can be regarded constant in regular duty cycles. Similar considerations with respect to the drag forces and train modeling are all based on similar thoughts, where the modeling will be generally precise enough when system dynamics can generally be ignored so that the computational efficiency can be achieved for potential online/offline applications. In the meantime, the highly dynamic efficiency mechanisms of the motors may also put further challenges on a precise mathematical modeling. For example, the motor efficiency is affected by the operation temperature and dynamic load conditions. This dynamic efficiency of motor would be beyond the scope of our current study in this paper. In general, we propose the adoption of a constant energy efficiency, in spite of the possibility of using a variable efficiency function in the model, due to two reasons: reducing model complexity to guarantee computation efficiency and considering motor efficiency as the average value for long-term operations.

　　Different from previous studies on full speed trajectory optimization, this paper is focused on a special problem, i.e., the adaptive partial speed trajectory optimization. The adaptive partial speed trajectory optimization is used to locate the speed trajectory with the minimum energy consumption in more general cases. The full speed trajectory considers cases with a zero initial speed and a final speed, which can otherwise be arbitrarily non-zero for partial speed trajectory optimization as shown in Figure 1. For example, when the vehicle is undergoing special operation sections, e.g., going through signal blocks or a junction area of railway networks ([4,5]), a partial speed trajectory optimization problem will be quickly adapted to special constraints imposed by signaling and multi-train operations. In the meantime, with a wider application of regenerative braking, partial speed trajectory optimization will be able to determine how much the most regenerated energy would be when a train is approaching the station. Similar to full speed trajectory optimization, partial speed trajectory optimization, to a large extent, could provide an optimal control strategy for the automatic train control (ATC) system or automatic train operation (ATO) system in online applications if the computational time is short enough. In this paper, we present adaptive partial speed trajectory optimization methods to highlight the adaptiveness of the proposed method ([6]), which is capable of dealing with any arbitrary initial and end speeds.

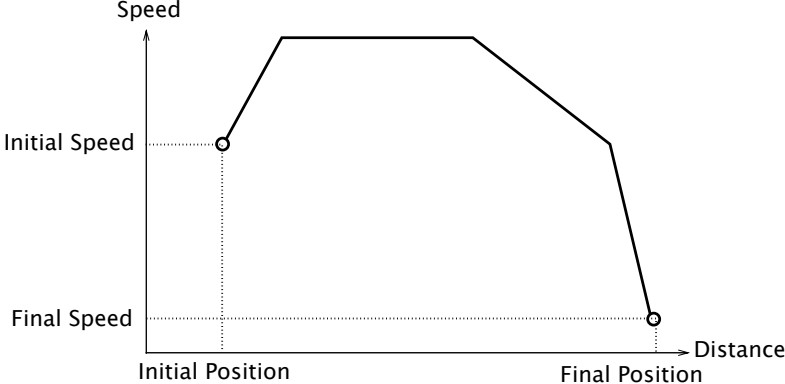

**Figure 1.** A typical optimal partial train speed trajectory between two non-zero speed points.

　　This paper aims to provide two different solutions to the adaptive partial speed optimization problem, i.e., optimal train control based on Pontryagin's maximum principle (PMP) and mixed-integer linear programming (MILP), to obtain the optimal partial speed trajectory considering motor efficiency. Based on the PMP, this paper provides explicit optimality analysis for nine different case scenarios

and summarizes the possible control strategies for each case. We also propose an intuitive numerical algorithm using the optimality analysis based on the PMP. Similar studies were once conducted in papers by [7,8]. On the other hand, a distance-based MILP model is proposed to solve the same optimization problem and provide optimization results to be compared and contrasted to the ones obtained from the PMP-based method. A thorough discussion based on the results from both methods will be conducted with an emphasis on the applicability and adaptivity of the mathematical programming method.

Compared to previous papers in the field, we aim to contribute to the literature in the following three aspects.

- First, from a viewpoint of train optimal control, we consider a special problem, i.e., the adaptive partial speed trajectory optimization problem with arbitrary initial and ending speeds. This optimization problem is of interest for more dynamic applications where the operation scenarios keep changing due to signaling and multi-train interactions. We present a detailed optimality analysis based on the PMP which is later used to design the optimization methods.
- Second, we propose two optimization methods: one based on the PMP as an indirect method and another based on MILP as a direct method, to solve the problem. Though both methods are popularly applied in train optimal control, such a comparative study has not yet been done. We develop an intuitive numerical algorithm to achieve the optimal speed trajectory in four typical case scenarios; meanwhile, we propose a distance-based MILP model to optimize the adaptive partial speed trajectory in the same scenarios with high modeling precisions.
- Third, in terms of traction technologies, we explicitly consider the motor efficiency from both traction and braking phases, rather than only the efficiency from regenerative braking phase or no efficiency considered compared to most existing studies, which has been demonstrated to impose significant impacts on optimal train speed trajectory.

This paper is arranged as follows. Section 1 provides an introduction on the research background and review on relevant literature. Section 2 gives a comprehensive optimality analysis on the adaptive partial speed trajectory optimization problem based on the PMP. Section 3 provides details about how to build the MILP model with a set of constraints to solve the problem. Section 4 demonstrates the results obtained from both PMP and MILP methods and provides comprehensive discussions based on the results. Section 6 draws the conclusion of this paper.

*1.1. Literature Review*

In Table 1, we list a number of selected papers in three main categories in terms of the main methods applied, i.e., optimal control based on the PMP, a heuristic algorithm, and mathematical programming.

The speed trajectory optimization problem has been tackled by a variety of methods in the past few decades. These methods can be generally categorized into optimal control, heuristic algorithm, and mathematical programming. Although the method based on optimal train control theory can be regarded as a type of mathematical programming due to its characteristics in minimizing the objective functional, we separate this method to recognize its important contribution on the area. In this paper, based on the optimality analysis using PMP, we propose an intuitive numerical algorithm and apply MILP to solve the adaptive partial speed trajectory optimization problem.

**Table 1.** Selected publications on train speed trajectory optimization.

| Methods | Publication | Algorithms/Theory | Multiple/Single Train(s) |
|---|---|---|---|
| Optimal control based on the PMP | [9] | PMP | Single |
| | [7] | PMP | Single |
| | [10] | PMP | Single |
| | [8] | PMP | Single |
| | [11,12] | PMP | Single |
| Heuristic algorithm | [13] | Genetic Algorithm | Single |
| | [14] | Genetic Algorithm, Ant Colony Optimization and Dynamic Programming | Single |
| | [15] | Genetic Algorithm | Multiple |
| | [16] | Brute force, Ant Colony Optimization and Genetic Algorithm | Multiple |
| | [17] | Genetic Algorithm | Multiple |
| | [18] | Genetic Algorithm | Single |
| Mathematical programming | [19] | Sequential Quadratical Programming | Single |
| | [20] | Pseudospectral method and MILP | Single |
| | [21] | Kuhn-Tucker Conditions | Multiple |
| | [22] | Bellman-ford Algorithm | Single |
| | [6] | MILP | Single |
| | [23] | Dynamic Programming | Single |
| | [4] | Pseudospectral method | Single |
| | [24] | Pseudospectral method | Multiple |
| | [25] | Genetic algorithm and Brute Force | Multiple |
| | [26] | Monte Carlo Simulation | Multiple |
| | This paper | MILP & PMP (Distance-based mathematical programming and PMP-based numerical algorithms) | Single/adaptive |

As early as 1968, the authors in [27] had applied the modern optimal train control theory based on the PMP in rail systems to optimize the train control strategy. Many similar studies on this area were conducted during 1970–2000. Some earlier works of significance can be found in papers by [28–31]. After 1990, the Scheduling and Control Group (SCG) at the University of South Australia developed the modern theory of train control in a collection of publications by [32–38]. These papers demonstrate that the study of optimal train control problems is taking more practical considerations with respect to the speed limit, traveling time, gradient, etc. and is targeting different train systems such as diesel freight trains and electric passenger trains.

The authors in [9] applied PMP to study the optimal control for both continuous and discrete control cases. Taking time as the independent variable in the model, the paper develops key equations to determine the optimal switch points based on the necessary conditions on an optimal strategy. The authors in [7] proposed the linking function to find the optimal trajectory and presented a numerical algorithm to solve the optimal train control problem with varying gradients and speed limits. The authors in [10] proposed an analysis method to deal with the switching point between each control strategy and, using a complementary variable, to meet the speed-limit constraints. The authors in [39] proposed a new local energy minimization principle to locate the critical switching points for a global optimal strategy on a track with steep gradients. [7,8] proposed applying a numerical algorithm to optimize the trajectory with a consideration of both varying gradient and speed limit. The authors in [40] studied the control operation during the varying gradient and the coasting operation on the slope and provided an analysis of a new local optimization principle. The authors in [11,12] summarized the key principles of optimal train control developed in the past few decades, in which different aspects of optimal train control problem have been summarized and discussed. These two papers are focused on the classic two-station train optimal control problem, where it is proved that an optimal strategy always exists in the proposed optimal train control model, and perturbation analysis

is used to show that the strategy is unique. In particular, regenerative braking has been taken into account, but the efficiency of motor during traction is not within the proposed research scope.

On the other hand, by modeling the train operations as a standard optimal control problem, researchers from Delft University of Technology and University of Leeds, proposed applying a pseudospectral algorithm and non-linear programming, as a direct method in comparison to an indirect method such as PMP, to solve the train speed trajectory optimization problem under a variety of engineering constraints. The studied cases can be well extended to multi-train problems and is able to take into account more complex signaling and operational constraints ([4,5,20,24,41]). The authors in [20] applies both a pseudospectral algorithm and MILP to solve the speed trajectory optimization problem with consideration of the passenger riding comfort. The problem is formulated as an optimal control problem and a comparative study is conducted between the two proposed methods. The authors in [4] presented a multiple-phase optimal control model for train trajectory optimization to be solved by a pseudospectral algorithm. The paper put particularly considered the operational (time and speed restrictions) and signal (signal aspects and automatic train protection) constraints for the trajectory optimization under delay and no-delay situations. The authors in [24] applied a pseudospectral method to optimize the multi-train optimization by formulating the problem as a multiple-phase optimal control problem. Multiple objectives including the minimization of energy consumption and train delays exist for the optimization to locate proper driving strategies. By modeling train operation in a closed-form model, the authors in [5] applied non-linear programming to address the train optimal control problem and the proposed methods have been further applied to formulate more complex optimal train control problems in scheduling and train following. In the above studies ([4,5,24]), undesirable fluctuation of speed trajectory and operations spanning across short durations are presented, which may hinder the direct applications of these methods if not properly addressed. Compared to these papers, our proposed methods have not yet considered multi-train operations and signaling constraints since our main objective of this paper is to target the partial speed trajectory optimization problem considering motor efficiency during both traction and braking with a detailed comparative study not yet conducted before with both a direct method (MILP) and an indirect method (PMP). Nevertheless, our proposed distance-based model solved by MILP is capable of an extension by taking into account similar signaling constraints by imposing linear constraints arising from an operational time window and multi-train operation constraints.

Apart from the optimal train control algorithm, the speed trajectory optimization problem has also been solved using different heuristic algorithms. As discussed in the book by [42], this is usually due to the fact that, with more practical constraints considered, many practical problems such as the speed trajectory optimization become an "NP-complete" problem. Heuristic algorithms, usually regarded as approximate algorithms, are applied to achieve near-optimal solutions with a reduced computational time. Typical heuristic algorithms include the ant colony algorithm (ACO), the genetic algorithm (GA), and the particle swarm algorithm. These algorithms are able to find the available solution with an acceptable cost arising from the consumed energy and elapsed time. The genetic algorithm (GA) is one of the most widely used methods in speed trajectory optimization ([13,15–18,43]). GA was applied to design a formal method to optimize the traction energy and to investigate the relationship between the journey time and energy consumption ([13]). The authors in [15] applied a GA to solve an optimization model targeting the minimization of net energy. In this model, both train operation and time tabling are considered so that the integrated model is able to take advantage of regenerative braking and speed trajectory optimization during the multi-train operations. As one of the advantages, different heuristic methods could be easily applied to solve the same optimization problem and make comparative studies on each algorithms.

The authors in [18] used a GA to solve the speed trajectory optimization problem with special consideration of regeneration braking so that the net energy could be reduced. The authors in [43] investigated the influence of the error of train positioning in optimal speed trajectory obtained using a GA. The speed trajectory optimization is based on a simple case with a single speed limit and

assumes that the operations of the train are designed in a preset sequence. In a recent paper by [17], a simulation-based GA was applied to solve a two-phase stochastic model considering the uncertain train mass to optimize the timetable and train speed profile. The optimization of the train speed profile was based on a simple assumption that the optimal train trajectory consists of maximum acceleration, coasting, and maximum deceleration. The authors in [25] proposed an integrated optimization model to simultaneously consider both timetabling and train trajectory for minimum energy consumption using GA and brute force methods. Based on Monte Carlo simulation, the authors in [26] presented an integrated optimization method to incorporate the train operation and electric network power flow.

Compared to the heuristic algorithms arising from the advances of computational intelligence, mathematical programming in more traditional optimization methods heavily relies on numerical iterations and mathematical modeling. With proper linearization and modeling of practical problems, such a method can also be robust and adaptive with guaranteed global optimality in convex cases. Many studies have also utilized mathematic programming to locate the optimal speed trajectory in different scenarios. The authors in [19] proposed a sequential quadratic programming (SQP) model to optimize the speed profile considering the charge and discharge of the on-board storage device. The authors in [14] achieved the optimal train speed trajectory in a discrete search space for a single train by using a GA, ACO, and dynamic programming using a distance–time–speed model, which has become a popular way of modeling in dealing with integrated optimization for both speed profiles and timetabling ([44]). The authors in [21] proposed an integrated model to minimize the energy in dynamic train scheduling and control in metro rail operations. In this paper, a convex optimization model is built to consider the train operation in terms of curve planning and scheduling in terms of journey time allocations simultaneously. Kuhn–Tucker conditions are applied to solve the optimization model to achieve a significant energy reduction. The authors in [45] applied an approximate dynamic programming to solve the train rescheduling problem in which speed trajectory was optimized in a discrete search space. The authors in [23] proposed a dynamic programming method to solve speed trajectory optimization using event-based decomposition to reduce the search space leading to significant computational time reduction. The authors in [44] proposed a unified modeling approach using space–time–speed to address the joint optimization problem for high-speed train timetables and speed profiles. Dynamic programming and Lagrangian relaxation were applied to address the power supply and safety constraints. However, motor characteristics, such as maximum power and maximum tractive effort, were not considered in the paper.

After 2014, the authors in [6,22,46,47] conducted a series of studies focused on the partial speed trajectory optimization problem. This problem arises from the question, "How much regenerative braking energy can be obtained from a braking procedure of train with a given travel distance and elapsed time?" The problem was first studied in the paper by [22] using the Bellman–Ford algorithm, i.e., a dynamic-programming-based graphical searching method, by modeling the braking speed trajectory in a discrete manner. An optimality analysis based on the PMP was conducted and high-level agreements between the analysis and optimization results were demonstrated. The authors in [46,47] proposed linear programming to solve monotonous speed trajectory during the regenerative braking process. The latter study applies a similar method in application of the eco-approach and departure for electric vehicles in signaling section areas. The authors in [6] applied MILP to address non-linear constraints arising from varying gradients along the braking route for partial speed trajectory optimization problems. By formulating the speed trajectory optimization as an MILP model, the authors in [48] tried to optimize the speed trajectory to achieve the minimum net energy. Further to this work, this paper develops and proposes an MILP model for adaptive partial speed trajectory optimization with considerations of motor efficiency during regenerative braking and traction.

In summary, train speed trajectory optimization has been long studied since the 1960s. A variety of methods such as optimal control theory, heuristic algorithms, and mathematical programming have been proposed in solving the problem. As discussed in two survey papers by [49,50], more studies were focused on the trajectory optimization in interconnected urban railway systems so that net

energy consumptions, passenger information, service qualities, and timetabling could be considered concurrently. As such, popularity has been gained on integrated models to realize a systematic integration of different railway planning and control aspects across decision layers in recent years. It has also been noted that many of the previous studies have not taken into account the energy efficiency during energy conversions and transmissions via electric motors and electric grids and other energy sub-systems, which is considered important in optimal train control strategies [26,51]. It is for this reason that this paper proposes the inclusion of the efficiency during both traction and regenerative braking operations, which is different from many previous papers, which only take efficiency in regenerative braking phases into account (if at all). We investigate the impact raised by these two efficiencies and set up an objective function to minimize the net electrical energy of the individual train but not the net mechanical energy only. Compared to previous studies on speed trajectory optimization between two adjacent stations, this paper tries to apply PMP and MILP to tackle an adaptive partial speed trajectory optimization problem between two arbitrary speed points, taking into account the efficiency of the motors during both traction and regenerative braking.

## 2. Optimality Analysis Based on Pontryagin's Maximum Principle

Given a unit mass, two state equations of train movement depending on distance are listed as follows:

$$\frac{dt}{dx} = \frac{1}{v} \tag{1a}$$

$$\frac{dv}{dx} = \frac{u_t f_t(v) - u_b f_b(v) - w_o(v) - g(x)}{v}. \tag{1b}$$

In Equations (1a) and (1b), $x$ is distance, $t$ is time, $v$ is train speed, $u_t$ and $u_f$ are the control variables ranging from 0 to 1, i.e., $u_t \in [0,1]$ and $u_b \in [0,1]$, $f_t(v)$ and $f_b(v)$ are the maximum electrical traction and braking forces related to $v$, $w_o(v)$ is the resistance force due to aerodynamic drags and rolling resistance and could be typically represented by a quadratic formula in respect to $v$, and $g(x)$ is the gradient depending on distance.

The objective function is to minimize the net electrical energy as shown below:

$$J = \int_0^S \left[ \frac{u_t f_t(v)}{\eta_t} - u_b f_b(v) \eta_b \right] dx \tag{2}$$

where $\eta_t$ and $\eta_b$ are the efficiency of electric motors ranging from 0 to 1 during the traction and braking operations. For traction operation, the electrical energy consumption can be regarded as the traction energy consumption divided by the efficiency $\eta_t$. For braking operation, the regenerated electrical energy can be regarded as the traction energy multiplied by the efficiency $\eta_b$.

Based on the PMP, the Hamiltonian is expressed as

$$
\begin{aligned}
H = & -\frac{u_t f_t(v)}{\eta_t} + u_b f_b(v) \eta_b \\
& + \frac{\lambda_1}{v} + \lambda_2 \frac{u_t f_t(v) - u_b f_b(v) - w_o(v) - g(x)}{v} \\
= & \left( \frac{\lambda_2}{v} - \frac{1}{\eta_t} \right) u_t f_t(v) + \left( \eta_b - \frac{\lambda_2}{v} \right) u_b f_b(v) \\
& + \frac{\lambda_1}{v} - \frac{\lambda_2}{v} \left[ w_o(v) + g(x) \right].
\end{aligned}
\tag{3}
$$

In Equation (3), $\lambda_1$ and $\lambda_2$ are the co-state variables. To simplify the form of Hamiltonian, we introduce another co-state variable $\theta$ to substitute $\lambda_2$, $\theta = \frac{\lambda_2}{v}$, and Equation (3) can then be transformed into Equation (4).

$$
\begin{aligned}
H = {} & \left(\theta - \frac{1}{\eta_t}\right) u_t f_t(v) + (\eta_b - \theta) u_b f_b(v) \\
& + \frac{\lambda_1}{v} - \theta[w_o(v) + g(x)].
\end{aligned}
$$

(4)

According to PMP, in order to minimize the objective function of Equation (2), the Hamiltonian Equation (4) needs to be maximized. Therefore, co-state variable $\theta$ can be divided into five different regions corresponding to different operations. The relation is summarized in Table 2.

**Table 2.** Different optimal control operation corresponding to different co-state variable and controlled variable.

| Co-State Variable | Optimal Control Operation | Controlled Variable |
|:---:|:---:|:---:|
| $\theta > \frac{1}{\eta_t}$ | Full traction | $u_t = 1$ and $u_b = 0$ |
| $\theta = \frac{1}{\eta_t}$ | Partial traction | $u_t \in [0,1]$ and $u_b = 0$ |
| $\eta_b < \theta < \frac{1}{\eta_t}$ | Coasting | $u_t$ and $u_b = 0$ |
| $\theta = \eta_b$ | Partial braking | $u_t = 0$ and $u_b \in [0,1]$ |
| $\theta < \eta_b$ | Full braking | $u_t = 0$ and $u_b = 1$ |

In our optimality analysis, we consider cases where no extreme gradient and speed limit exists. Extreme gradients exist when the following two inequality applies.

$$
f_t(v) - w_0(v) - g(x) < 0
$$

(5)

$$
-f_b(v) - w_0(v) - g(x) > 0.
$$

(6)

Based on the PMP, the following two equations should be satisfied.

$$
\frac{d\lambda_1}{dx} = -\frac{\partial H}{\partial t}
$$

(7)

$$
\frac{d\lambda_2}{dx} = -\frac{\partial H}{\partial v}.
$$

(8)

Since the Hamiltonian Equation (3) is not directly related to $t$, $\frac{d\lambda_1}{dx} = 0$, and this leads to a constant value of $\lambda_1$. In addition, Equation (8) can be further presented as

$$
\begin{aligned}
d\lambda_2 = {} & -\frac{\partial H}{\partial v}dx \\
= {} & \left[\frac{u_t}{\eta_t}f_t'(v) - u_b\eta_b f_b'(v) + \frac{\lambda_1}{v^2}\right. \\
& - \lambda_2 \frac{u_t f_t'(v) - w_o'(v) - u_b f_b'(v)}{v} \\
& \left. + \lambda_2 \frac{u_t f_t(v) - w_o(v) - u_b f_b(v) - g(x)}{v^2}\right] dx \\
= {} & \left[\left(\frac{1}{\eta_t} - \frac{\lambda_2}{v}\right) u_t f_t'(v) + \left(\frac{\lambda_2}{v} - \eta_b\right) u_b f_b'(v) \right. \\
& \left. + \frac{\lambda_2}{v}w_o'(v) + \frac{\lambda_1}{v^2} + \frac{\lambda_2 dv}{vdx}\right] dx.
\end{aligned}
$$

(9)

The state function of Equation (1b) can be rewritten with the following relation:

$$\frac{dv}{vdx} = \frac{u_t f_t(v) - w_o(v) - u_b f_b(v) - g(x)}{v^2}.$$

(10)

Assume a new costate variable $\lambda_2 = \theta v$, and we have the following:

$$d\lambda_2 = vd\theta + \theta dv.$$

(11)

Combining Equations (9) and (11), we can get the form of the $d\theta$ expressed by $\theta$.

$$d\theta = \left[ \frac{1/\eta_t - \theta}{v} u_t f_t'(v) + \frac{\theta - \eta_b}{v} u_b f_b'(v) \right.$$
$$\left. + \frac{\theta w_o'(v)}{v} + \frac{\lambda_1}{v^3} \right] dx.$$

(12)

Without the influence of the speed limit, $\theta$ is continuous, and $\frac{d\theta}{dx}$ is differentiable along the track.

Depending on different gradients, two cruising operations exist using either partial traction or partial braking. The partial traction is analyzed as follows. During the partial traction, $\theta = \frac{1}{\eta_t}$, $u_b = 0$, and $u_t \in [0, 1]$, as shown in Table 2.

Equation (12) is transformed as follows:

$$\frac{d\theta}{dx} = \frac{w_o'(v)}{\eta_t v} + \frac{\lambda_1}{v^3} = 0.$$

(13)

We rearrange Equation (13), and it leads to

$$-\lambda_1 = \frac{v^2 w_o'(v)}{\eta_t}.$$

(14)

Since $\lambda_1$ is a constant, we can reasonably assume there is a constant $V_{tc}$ corresponding a constant $\lambda_1$.

$$-\lambda_1 = \frac{V_{tc}^2 w_o'(V_{tc})}{\eta_t}.$$

(15)

We then use $\varphi(v)$ to denote the term $v^2 w_o'(v)$, where $\varphi(v)$ is a monotonously increasing function of $v$.

We substitute it into Equation (15) and obtain

$$\frac{1}{\eta_t} \varphi(V_{tc}) = \frac{V_{tc}^2 w_o'(V_{tc})}{\eta_t} = -\lambda_1.$$

(16)

Similarly, during the partial braking operation, we can deduce the relation between $\lambda_1$ and $\varphi(V_{bc})$.

$$\eta_b \varphi(V_{bc}) = \eta_b V_{bc}^2 w_o'(V_{bc}) = -\lambda_1.$$

(17)

With the motor efficiency $\eta_b$ and $\eta_t$ ranging from 0 to 100%, we can derive $V_{bc} > V_{tc}$.

If the speed value of train and co-state variable are given, the evolutionary trend of $\theta$ can be deducted. Based on the current speed of the train in relation to $V_{bc}$ and $V_{tc}$, we make the following analysis in various cases.

## 2.1. Optimality Analysis for High Train Speed

### 2.1.1. Case 1: $v > V_{bc}$, $\theta < \eta_b$

We consider a case with $V_{bc}$ by using partial braking, the speed $v > V_{bc}$, and the co-state variable $\theta < \eta_b$. Equation (12) can be transformed into

$$
\begin{aligned}
\frac{d\theta}{dx} &= (\theta - \eta_b)\frac{f'_b(v)}{v} + \theta\frac{w'_o(v)}{v} + \frac{\lambda_1}{v^3} \\
&= \frac{1}{v^3}\left[v^2(\theta - \eta_b)f'_b(v) + \theta\varphi(v) - \eta_b\varphi(V_{bc})\right] \\
&= \frac{1}{v^3}\{\theta[\phi(v) + \varphi(v)] - \eta_b[\varphi(V_{bc}) + \phi(v)]\}.
\end{aligned}
\tag{18}
$$

In Equation (18), we use $\phi(v)$ to denote $v^2 f'_b(v)$. We set $\frac{d\theta}{dx} = 0$. Assume that there is one $\theta_x$ resulting in $\frac{d\theta}{dx} = 0$, and we have

$$
\theta_{x1} = \eta_b\frac{\varphi(V_{bc}) + \phi(v)}{\varphi(v) + \phi(v)}.
\tag{19}
$$

To determine the relation between $\theta_{x1}$ and $\eta_b$, we first consider the sign of the $\varphi(v) + \phi(v)$. We introduce an equation $\zeta(v) = f'_b(v) + w'_o(v)$. Considering the traction characteristics of electric motors, the braking and traction effort at low speed can be regarded as constant. This gives $f'_b(v) = 0$. In addition, $w'_o(v)$ is positive and this leads to $\zeta(0) > 0$. We then can calculate $\zeta'(v)$ based on the characteristics of $f_b(v)$ and $w_o$ ($f_b(v)$ is considered to be in proportion to $\frac{1}{v}$, and $w_o(v)$ is in a quadratic form of $v$.), and it is derived that $\zeta'(v) > 0$. Therefore, we conclude that $\zeta(v) > 0$, and this leads to $\varphi(v) + \phi(v) > 0$. Due to the monotonicity of $\phi(v)$ and $\varphi(v)$, we can determine that $\theta_{x1} < \eta_b$.

With a given $v$, $\frac{d\theta}{dx}$ in Equation (18) is positively linear to $\theta$. A schematic to illustrate the relationship between $\theta$ and $\frac{d\theta}{dx}$ is shown in Figure 2.

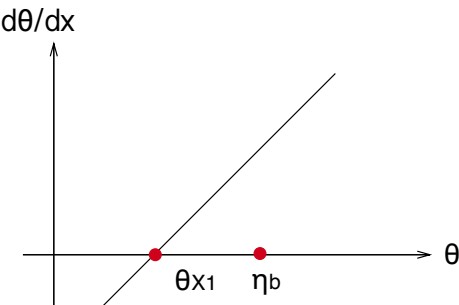

**Figure 2.** $\frac{d\theta}{dx}$ vs. $\theta$ in Case 1.

We can make the following remarks.

- **Case 1.1:** If $\theta > \theta_{x1}$, $\frac{d\theta}{dx} > 0$. With $v > V_{bc}$ and $\theta_{x1} < \theta < \eta_b$, the optimal evolution of $\theta$ will be increasing and $v$ will be decreasing until it reaches the condition $v = V_{bc}$ and $\theta = \eta_b$.
- **Case 1.2:** If $\theta < \theta_{x1}$, $\frac{d\theta}{dx} < 0$. Therefore, with $v > V_{bc}$ and $\theta < \theta_{x1}$, $\theta$ will be decreasing and $v$ will be decreasing until it reaches the final state.

2.1.2. Case 2: $v > V_{bc}$, $\eta_b < \theta < \frac{1}{\eta_t}$

There exists a state with $V_{bc}$, $v > V_{bc}$, and $\eta_b < \theta < \frac{1}{\eta_t}$. Equation (12) can be transformed into

$$
\begin{aligned}
\frac{d\theta}{dx} &= \theta \frac{w_o'(v)}{v} + \frac{\lambda_1}{v^3} \\
&= \frac{1}{v^3} \left[ \theta \varphi(v) - \eta_b \varphi(V_{bc}) \right].
\end{aligned}
\tag{20}
$$

$\eta_b < \theta < \frac{1}{\eta_t}$, $v > V_{bc}$, and $\varphi(v)$ is a monotonously increasing function of $v$. These conditions can guarantee $\frac{d\theta}{dx} > 0$.

As a result, if there is a coasting operation with $\eta_b < \theta < \frac{1}{\eta_t}$, this will lead to $\frac{d\theta}{dx} > 0$. $\theta$ will be increasing and $v$ will be decreasing until it reaches the condition $v = V_{bc}$ and $\theta = \frac{1}{\eta_t}$.

2.1.3. Case 3: $v > V_{bc}$, $\theta > \frac{1}{\eta_t}$

If there exists a constant cruising speed $V_{bc}$, $v > V_{bc}$, and $\theta > \frac{1}{\eta_t}$, Equation (12) can be transformed into

$$
\begin{aligned}
\frac{d\theta}{dx} &= \frac{1/\eta_t - \theta}{v} f_t'(v) + \theta \frac{w_o'(v)}{v} + \frac{\lambda_1}{v^3} \\
&= \frac{1}{v^3} \left[ v^2 (1/\eta_t - \theta) f_t'(v) + \theta \varphi(v) - \frac{1}{\eta_t} \varphi(V_{tc}) \right].
\end{aligned}
\tag{21}
$$

With the conditions that $\theta > \frac{1}{\eta_t}$, $v > V_{bc}$, and $\varphi(v)$ is a monotonously increasing function of $v$, $f_t'(v)$ is negative. We can determine the term $\theta \varphi(v) - \frac{1}{\eta_t} \varphi(V_{tc}) > 0$. This can guarantee $\frac{d\theta}{dx} > 0$.

As a result, if there is an acceleration operation with $\theta > \frac{1}{\eta_t}$, $v > V_{bc}$, $\theta$ will be increasing and $v$ will be increasing. In this case, $\theta > \frac{1}{\eta_t}$ along $x$. The corresponding optimal operation will be full traction until it arrives at the final speed.

*2.2. Optimality Analysis for Medium Train Speed*

2.2.1. Case 4: $V_{tc} < v < V_{bc}$, $\theta < \eta_b$

There are two constant cruising speeds $V_{bc}$ and $V_{tc}$. The train speed $V_{tc} < v < V_{bc}$ and $\theta < \eta_b$. Equation (12) can be transformed into Equation (22):

$$
\begin{aligned}
\frac{d\theta}{dx} &= (\theta - \eta_b) \frac{f_b'(v)}{v} + \theta \frac{w_o'(v)}{v} + \frac{\lambda_1}{v^3} \\
&= \frac{1}{v^3} \left[ v^2 (\theta - \eta_b) f_b'(v) + \theta \varphi(v) - \eta_b \varphi(V_{bc}) \right] \\
&= \frac{1}{v^3} \{ \theta [\phi(v) + \varphi(v)] - \eta_b [\varphi(V_{bc}) + \phi(v)] \}.
\end{aligned}
\tag{22}
$$

Similar to Case 1, we assume that there is one $\theta_{x2}$ leading to $\frac{d\theta}{dx} = 0$, so we have Equation (23). Using a similar deduction as in Case 1, it can be shown that $\theta_{x2} > \eta_b$. With a given positive $v$, we are able to present the relationship between $\frac{d\theta}{dx}$, and $\theta$ in positive linearity as shown in Figure 3. In this case, we have $\frac{d\theta}{dx} < 0$. Therefore, $\theta$ and $v$ will be decreasing until it reaches the final speed.

$$
\theta_{x2} = \eta_b \frac{\varphi(V_{bc}) + \phi(v)}{\varphi(v) + \phi(v)}.
\tag{23}
$$

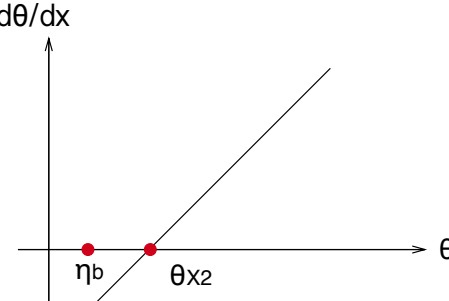

**Figure 3.** $\frac{d\theta}{dx}$ vs. $\theta$ in Case 4.

2.2.2. Case 5: $V_{tc} < v < V_{bc}, \eta_b < \theta < \frac{1}{\eta_t}$

In this case, the train speed is maintained by the coasting operation, and the train speed falls between two cruising speeds: $V_{tc} < v < V_{bc}$ and $\eta_b < \theta < \frac{1}{\eta_t}$. Equation (12) can be transformed into Equation (24) as follows:

$$
\begin{aligned}
\frac{d\theta}{dx} &= \theta \frac{w_o'(v)}{v} + \frac{\lambda_1}{v^3} \\
&= \frac{1}{v^3}\left[\theta\varphi(v) - \eta_b\varphi(V_{bc})\right] \\
&= \frac{1}{v^3}\left[\theta\varphi(v) - \frac{1}{\eta_t}\varphi(V_{tc})\right].
\end{aligned}
\tag{24}
$$

With $\eta_b < \theta < \frac{1}{\eta_t}$, $V_{tc} < v < V_{bc}$, and $\varphi(v)$ being a monotonously increasing function of $v$, we are able to calculate the value of $\theta_{x3}$, leading to $\frac{d\theta}{dx} = 0$.

$$
\theta_{x3} = \eta_b \frac{\varphi(V_{bc})}{\varphi(v)} = \frac{1}{\eta_t}\frac{\varphi(V_{tc})}{\varphi(v)}.
\tag{25}
$$

Equation (25) shows $\eta_b < \theta_{x3} < \frac{1}{\eta_t}$. As shown in Figure 4, there are two possible cases.

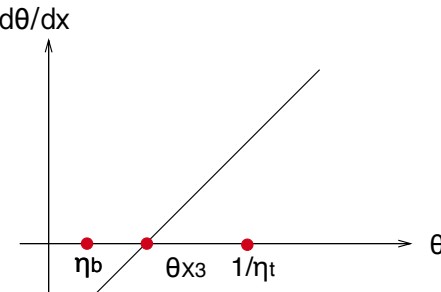

**Figure 4.** $\frac{d\theta}{dx}$ in Case 5.1 and Case 5.2.

- **Case 5.1:** $\theta > \theta_{x3}$ leads to $\frac{d\theta}{dx} > 0$. As a result, if $\eta_b < \theta < \frac{1}{\eta_t}$, $V_{tc} < v < V_{bc}$, $\theta$ will be increasing and $v$ will be decreasing until it reaches the condition $\theta = \frac{1}{\eta_t}$.
- **Case 5.2:** $\theta < \theta_{x3}$ leads to $\frac{d\theta}{dx} < 0$. Therefore, with $V_{tc} < v < V_{bc}$ and $\eta_b < \theta < \frac{1}{\eta_t}$, $\theta < \theta_{x3}$, $\theta$ will be decreasing and $v$ will be decreasing until it reaches the condition $\theta = \eta_b$.

### 2.2.3. Case 6: $V_{tc} < v < V_{bc}, \theta > \frac{1}{\eta_t}$

The train is with a speed $V_{tc} < v < V_{bc}$ and co-state variable $\theta > \frac{1}{\eta_t}$. Equation (12) can be transformed into Equation (26):

$$
\begin{aligned}
\frac{d\theta}{dx} &= \frac{1/\eta_t - \theta}{v} f_t'(v) + \theta \frac{w_o'(v)}{v} + \frac{\lambda_1}{v^3} \\
&= \frac{1}{v^3} \left[ v^2(1/\eta_t - \theta) f_t'(v) + \theta \varphi(v) - \frac{1}{\eta_t} \varphi(V_{tc}) \right] \\
&= \frac{1}{v^3} \{ \theta[\phi(v) - \varphi(v)] - \frac{1}{\eta_t}[\phi(v) - \varphi(V_{tc})] \}.
\end{aligned}
\tag{26}
$$

Similar to Case 1, we assume that there is one $\theta_{x4}$ leading to $\frac{d\theta}{dx} = 0$, so we have Equation (27). Using a similar deduction as in Case 1, it can be shown that $\theta_{x4} < \frac{1}{\eta_t}$. With a given positive $v$, the relationship between $\frac{d\theta}{dx}$ and $\theta$ is positively linear as shown in Figure 5. With $V_{tc} < v < V_{bc}$ and $\theta > \frac{1}{\eta_t}$, we can conclude that $\frac{d\theta}{dx} > 0$. As a result, both $\theta$ and $v$ will be decreasing until it reaches the final speed.

$$
\theta_{x4} = \frac{1}{\eta_t} \frac{\phi(v) - \varphi(V_{tc})}{\phi(v) - \varphi(v)}.
\tag{27}
$$

With $V_{tc} < v < V_{bc}$ and $\theta > \frac{1}{\eta_t}$, the optimal evolution of $\theta$ and $v$ will be increasing until it reaches the condition $v = V_{bc}$.

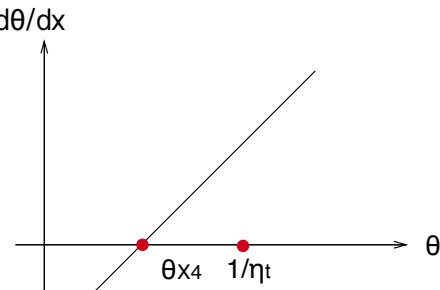

**Figure 5.** $\frac{d\theta}{dx}$ vs. $\theta$ in Case 6.

### 2.3. Optimality Analysis for Low Train Speed

### 2.3.1. Case 7: $v < V_{tc}, \theta < \eta_b$

With $V_{tc}, v < V_{tc}$ and $\theta < \eta_b$, Equation (12) can be transformed into Equation (28):

$$
\begin{aligned}
\frac{d\theta}{dx} &= (\theta - \eta_b) \frac{f_b'(v)}{v} + \theta \frac{w_o'(v)}{v} + \frac{\lambda_1}{v^3} \\
&= \frac{1}{v^3} \left[ v^2(\theta - \eta_b) f_b'(v) + \theta \varphi(v) - \eta_b \varphi(V_{bc}) \right] \\
&= \frac{1}{v^3} \{ \theta[\phi(v) + \varphi(v)] - \eta_b[\varphi(V_{bc}) + \phi(v)] \}.
\end{aligned}
\tag{28}
$$

Similar to Case 1, we assume that there is one $\theta_{x5}$, which leads to the zero derivative of $\theta$ and we have

$$
\theta_{x5} = \eta_b \frac{\varphi(V_{bc}) + \phi(v)}{\varphi(v) + \phi(v)}.
\tag{29}
$$

As shown in Figure 6, we find that $\theta_{x5} > \eta_b$ and with a given $v$, it is guaranteed that $\frac{d\theta}{dx} < 0$. As a result, we demonstrate that, if $\theta < \eta_b, v < V_{tc}$, we have $\frac{d\theta}{dx} < 0$. The optimal $\theta$ and $v$ will keep decreasing until it reaches a final speed.

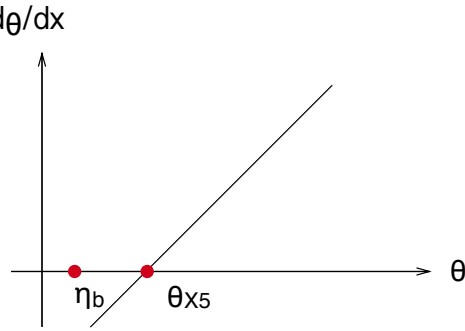

**Figure 6.** $\frac{d\theta}{dx}$ of Case 7.

### 2.3.2. Case 8: $v < V_{tc}$, $\eta_b < \theta < \frac{1}{\eta_t}$

We consider a case with $v < V_{tc}$ and $\eta_b < \theta < \frac{1}{\eta_t}$. Equation (12) can be transformed into

$$
\begin{aligned}
\frac{d\theta}{dx} &= \theta \frac{w_o^{'}(v)}{v} + \frac{\lambda_1}{v^3} \\
&= \frac{1}{v^3}\left[\theta\varphi(v) - \frac{1}{\eta_t}\varphi(V_{tc})\right].
\end{aligned}
\tag{30}
$$

With $\eta_b < \theta < \frac{1}{\eta_t}$, $v > V_{tc}$ and $\varphi(v)$ being a monotonously increasing function of $v$, we have $\frac{d\theta}{dx} < 0$.

As a result, we demonstrate that in Case 8, $\frac{d\theta}{dx} < 0$. The optimal $\theta$ will be decreasing and $v$ will be decreasing until it reaches the condition $\theta = \eta_b$.

### 2.3.3. Case 9: $v < V_{tc}$, $\theta > \frac{1}{\eta_t}$

In Case 9, we consider a case with a constant cruising speed $V_{tc}$, $v < V_{tc}$, and $\theta > \frac{1}{\eta_t}$. Equation (12) can be transformed into Equation (31):

$$
\begin{aligned}
\frac{d\theta}{dx} &= \frac{1/\eta_t - \theta}{v}f_t^{'}(v) + \theta\frac{w_o^{'}(v)}{v} + \frac{\lambda_1}{v^3} \\
&= \frac{1}{v^3}\left[v^2(1/\eta_t - \theta)f_t^{'}(v) + \theta\varphi(v) - \frac{1}{\eta_t}\varphi(V_{tc})\right] \\
&= \frac{1}{v^3}\{\theta[\phi(v) - \varphi(v)] - \frac{1}{\eta_t}[\phi(v) - \varphi(V_{tc})]\}.
\end{aligned}
\tag{31}
$$

Similar to Case 1, we assume that there is one $\theta_{x6}$ leading to the zero derivative of $\theta$, so we have Equation (32).

$$
\theta_{x6} = \frac{1}{\eta_t}\frac{\phi(v) - \varphi(V_{tc})}{\phi(v) - \varphi(v)}.
\tag{32}
$$

Using a similar deduction as in Case 1 by considering the characteristics of both $\phi(v)$ and $\varphi(v)$, it can be shown that $\theta_{x5} > \frac{1}{\eta_t}$. With a given positive $v$, the relationship between $\frac{d\theta}{dx}$ and $\theta$ is positively linear as shown in Figure 7. With $v < V_{tc}$ and $\theta > \frac{1}{\eta_t}$, we make the following two remarks.

- **Case 9.1:** If $\theta > \theta_{x6}$, $\frac{d\theta}{dx} > 0$, the optimal $\theta$ and $v$ will be increasing via acceleration until it reaches the final speed.
- **Case 9.2:** If $\theta < \theta_{x6}$, $\frac{d\theta}{dx} < 0$, the optimal $\theta$ will be decreasing and $v$ will be increase until it reaches the condition $v = V_{tc}$ and $\theta = \frac{1}{\eta_t}$.

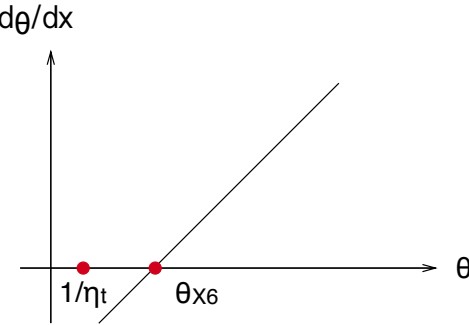

**Figure 7.** $\frac{d\theta}{dx}$ of Case 9.1 and Case 9.2.

### 2.4. Summary of Optimality Analysis and the Developed Numerical Algorithms

Based on the above discussions, we can summarize the trend of $\theta$ and the corresponding train operations from different values in respect to $\eta_b$, $\frac{1}{\eta_t}$, $V_{tc}$, and $V_{bc}$. The information is summarized in Table 3.

**Table 3.** Summary of $\frac{d\theta}{dx}$ and the corresponding optimal train operations.

| Case | $v$ | $\theta$ | $\frac{d\theta}{dx}$ | **Optimal Train Operation** |
|---|---|---|---|---|
| 1.1 | $v > V_{bc}$ | $\theta_{x1} < \theta < \eta_b$ | $\frac{d\theta}{dx} > 0$ | Full Braking $\rightarrow$ Partial Braking or Full Braking $\rightarrow$ Coasting |
| 1.2 | $v > V_{bc}$ | $\theta < \theta_{x1}$ | $\frac{d\theta}{dx} < 0$ | Full Braking |
| 2 | $v > V_{bc}$ | $\eta_b < \theta < \frac{1}{\eta_t}$ | $\frac{d\theta}{dx} > 0$ | Coasting $\rightarrow$ Full Traction or Partial Traction |
| 3 | $v > V_{bc}$ | $\theta > \frac{1}{\eta_t}$ | $\frac{d\theta}{dx} > 0$ | Full Traction |
| 4 | $V_{tc} < v < V_{bc}$ | $\theta < \eta_b$ | $\frac{d\theta}{dx} < 0$ | Full Braking |
| 5.1 | $V_{tc} < v < V_{bc}$ | $\eta_b < \theta < \theta_{x3}$ | $\frac{d\theta}{dx} < 0$ | Coasting $\rightarrow$ Partial Braking |
| 5.2 | $V_{tc} < v < V_{bc}$ | $\theta_{x3} < \theta < \frac{1}{\eta_t}$ | $\frac{d\theta}{dx} > 0$ | Coasting $\rightarrow$ Partial Traction |
| 6 | $V_{tc} < v < V_{bc}$ | $\theta > \frac{1}{\eta_t}$ | $\frac{d\theta}{dx} > 0$ | Full Traction |
| 7 | $v < V_{tc}$ | $\theta < \eta_b$ | $\frac{d\theta}{dx} < 0$ | Full Braking |
| 8 | $v < V_{tc}$ | $\eta_b < \theta < \frac{1}{\eta_t}$ | $\frac{d\theta}{dx} < 0$ | Coasting $\rightarrow$ Full Braking |
| 9.1 | $v < V_{tc}$ | $\theta > \theta_{x6}$ | $\frac{d\theta}{dx} > 0$ | Full Traction |
| 9.2 | $v < V_{tc}$ | $\theta_{x6} > \theta > \frac{1}{\eta_t}$ | $\frac{d\theta}{dx} < 0$ | Full Traction $\rightarrow$ Partial Traction |

In this paper, we propose two numerical algorithms based on the PMP to find the optimal speed trajectory. The obtained results will be compared and contrasted with the one obtained using the MILP mathematical programming method. In our proposed numerical algorithms, we are to apply linear iteration using the corresponding co-state equations in each case and use a simple intuitive search method to search for the initial and final co-state variable values, or the initial and final distances. The linear iteration is based on a sufficiently small distance step $\Delta d$ ([52]).

We propose Algorithm 1 to link the initial speed $v_0$ and the final speed $v_t$ to the cruising speeds $v_{tc}$ or $v_{bc}$. Once the initial and final co-state variable is located, the entire speed trajectory can be determined. A schematic for the scenario setup is shown in Figure 8.

---

**Algorithm 1** To achieve the optimal speed trajectory between a given initial speed $v_0$ and final speed $v_t$ to cruising speeds

---

Initialize the initial $\theta_1$ based on Table 3 and the initial train speed $v_1$ with $v_0$ or $v_t$
Allocate $\theta_N$ and $s_N$ with a value of zero
Define a sufficiently small number $\epsilon$
**while** $|\theta_N - \eta_b| > \epsilon$ for cruising cases using braking or $|\theta_N - 1/\eta_t| > \epsilon$ for cruising cases using

traction or $s_N$ is outside the cruising section range **do**

    Re-allocate $\theta_N$ and $s_N$ with a value of zero
    **repeat**

        Based on the current value of co-state variable $\theta$, speed $v$ and distance $s$, referring to Table 3 and

        Equation (12) to calculate the value of $v$ and $\theta$ for the next step within a small distance interval

        $\Delta d$. Apply backwards calculations for $v_t$ case.
        Update the current distance with a small distance interval
    **until** The train speed $v$ arrives at the cruising speed; the current corresponding $\theta$ to be assigned to

    $\theta_N$ and the current distance to be assigned to $s_N$
    Generate a new $\theta_1$ by $\theta_1 = \theta_1 + \Delta\theta$ where $\Delta\theta$ is the search step.
**end while**

---

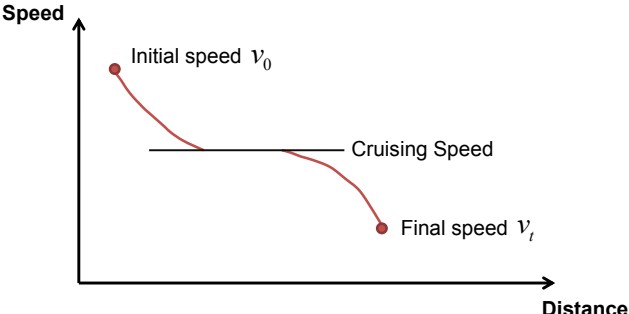

**Figure 8.** Schematic for the scenario setup in Algorithm 1.

On the other hand, if there exists two cruising speeds that need to be joined together. In this case, the initial $\theta_0$ and final $\theta_t$ are both known, and we propose Algorithm 2 as below to obtain the initial and final locations for the joining process. A schematic for the scenario setup has been shown in Figure 9.

In the proposed case studies in this paper, Algorithm 1 is applied to achieve the results in Scenarios 1–3, and Algorithm 2 is applied to achieve the results in Scenario 4 in Section 4. According to optimal train control theory ([7,11,12]), as long as these two algorithms are able to connect the initial speed, the final speed, and all cruising speeds, the obtained speed trajectory can be considered as the speed trajectory satisfying the necessary conditions of optimal speed trajectory. The next section will verify the optimality based on the optimized results using the MILP algorithm.

---

**Algorithm 2** To achieve the optimal speed trajectory and its initial location and final locations between two cruising speeds

---

Initialize the initial location $s_1$ of train within the first section with an initial value of $\theta_1$ with a small variable from $\eta_b$ or $1/\eta_t$ to ensure a valid evolution;

Allocate $\theta_N$ with a value of zero

Define a sufficiently small number $\epsilon$

**while** $|\theta_N - \eta_b| > \epsilon$ or $|\theta_N - 1/\eta_t| > \epsilon$ or $s_N$ is outside the range of the next cruising section **do**

  Re-allocate $\theta_N$ and $s_N$ with a value of zero.

  **repeat**

    Based on the current value of co-state variable $\theta$, speed $v$ and distance $s$, referring to Table 3 and Equation (12) to calculate the value of $v$ and $\theta$ for the next step within a small distance interval $\Delta d$

    Update the current distance with the additional small distance interval

  **until** The train speed $v$ arrives at the next cruising speed or the train location has gone beyond the next cruising section

  The current corresponding $\theta$ to be assigned to $\theta_N$ and the current distance to be assigned to $s_N$

  Generate a new beginning location $s_1$ by $s_1 = s_1 + \Delta s$ where $\Delta s$ is the search step.

**end while**

---

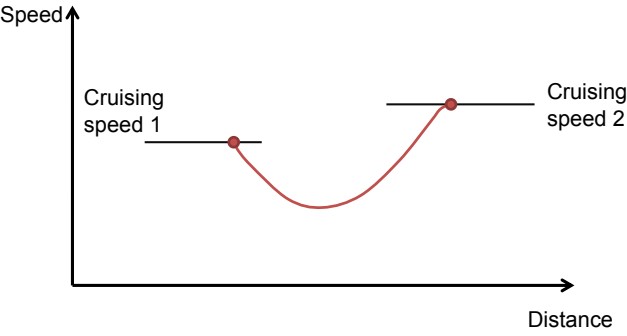

**Figure 9.** Schematic for the scenario setup in Algorithm 2.

## 3. Distance-Based Partial Speed Trajectory Optimization Model Using Mixed Integer Linear Programming

In the MILP model, the entire journey is divided into a series of distance sections. Compared to other methods, MILP takes advantage in shorter calculation time by giving a reasonable sacrifice in model precision. On the other hand, MILP also has the high flexibility and efficiency in simulation and modeling as a programming method. In the previous studies on this area proposed by [6,46,47], the MILP model could obtain the optimal partial speed trajectory with high efficiency and robustness.

However, there are still some limitations for the previously proposed MILP models ([6,47]). One of the main limitations is due to the fundamental assumption on the monotonicity of the speed trajectory during braking or acceleration processes. This issue can be addressed by proposing a distance-based model where the speed of train is not necessarily monotonous. The determinant variable will be changed from the distance interval to the speed/kinetic energy at each designated location, i.e., $v_0, v_1, \cdots, v_N$, as shown in Figure 10. The proposed MILP model can deal with more practical scenarios between any two arbitrary speed points with a given distance and traveling time. The result obtained by the MILP model can be verified by the optimality analysis of PMP. The speed trajectory is discretized into a series of sections. Therefore, each section can represent a part of the speed trajectory, and the whole speed trajectory could be obtained by connecting each section together.



In this model, maximum power, traction/braking efforts and acceleration, and motor efficiency are taken into account by incorporating a set of linear constraints.

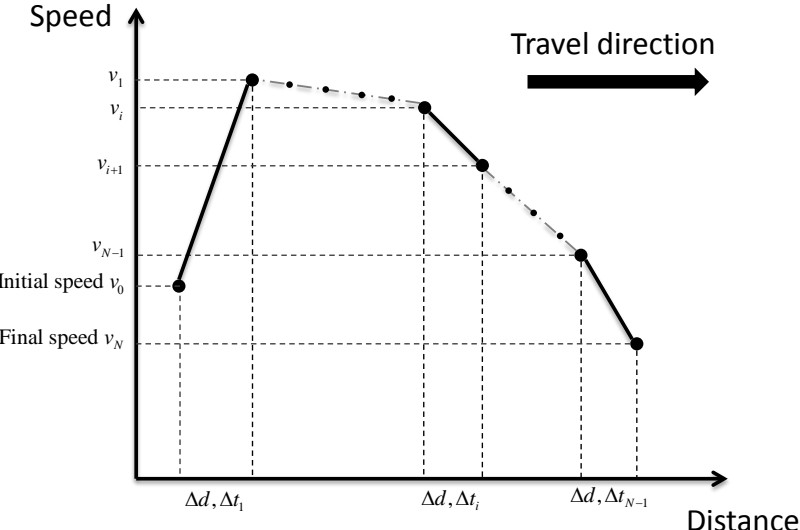

**Figure 10.** Discretized speed trajectory model solved by mixed-integer linear programming (MILP).

In Figure 10, the whole speed trajectory is discretized into a series of sections with a distance of $\Delta d$. $\Delta d$ should be sufficiently small to ensure high calculation precision. Each section could represent a part of the trajectory which contains the traveled distance, elapsed time, and consumed energy. N is assumed to represent the number of sections. The distance for each section is $\Delta d$, and the electrical energy for each section are referred to as $E_1, E_2, \cdots, E_i, \cdots, E_N$, the objective function of the model is the sum of the electrical energy input/output from the motor in each section as shown in Equation (33). The speed points are represented by the square of speed as $v_0^2, v_2^2, \cdots, v_i^2, \cdots, v_N^2$, which are the determinant variables. Some non-linear constraints such as traveling time will be represented using a piecewise linear (PWL) relationship between time and speed point. The reason for using the square of speed as the determinant variable itself is to reduce the number of non-linear variables in the model.

$$f = \sum_{i=1}^{N-1} E_i. \tag{33}$$

In each section of the model, the vehicle is regarded moving with a constant acceleration $a_i$ between two adjacent speed points $v_i$ and $v_{i+1}$. In practice, there are many constraints about each variable in the model. If the maximum acceleration or deceleration rate for the vehicle is assumed as $A_{max}$, the square of speed for each section should satisfy the constraint in Equation (34).

$$-A_{max} \le \frac{v_{i+1}^2 - v_i^2}{2\Delta d} \le A_{max}. \tag{34}$$

In order to obtain the drag force applied to the vehicle, the Davis equation is demonstrated in a quadratic form as shown in Equation (35). The $v_{i,ave}$ is referred to as the average speed of the train for each section.

$$F_{i,drag} = A + Bv_{i,ave} + Cv_{i,ave}^2 \tag{35}$$

where $A$, $B$, and $C$ are the Davis coefficients.

Because the optimal variables are assumed to be $v_i^2$, the $v_{i,ave}$ could be obtained by PWL as shown in Equations (36)–(40). PWL is a kind of method to represent the nonlinear variable in a linear relationship. $\alpha$ is special ordered set Type 2 (SOS2) ([53]). By definition, SOS2 is a set of variables

where at most two variables may be nonzero. If two variables are nonzero, they must be adjacent in the set. In this model, it will consider the vehicle operating on a speed level ranging from 1 to 50 m/s.

$$1 = \alpha_{i,1} + \alpha_{i,2} + \cdots + \alpha_{i,j} + \cdots + \alpha_{i,50} \tag{36}$$

$$0 \leq \alpha_{i,j} \leq 1 \quad j = 1, 2, \cdots, 50 \tag{37}$$

$$v_i'^2 = 50^2 \alpha_{i,1} + (51 - 2)^2 \alpha_{i,2} + \cdots + (51 - j)^2 \alpha_{i,j} + \cdots + 1^2 \alpha_{i,50} \tag{38}$$

$$v_i = 50\alpha_{i,1} + 49\alpha_{i,2} + \cdots + (51 - j)\alpha_{i,j} + \cdots + 1\alpha_{i,50} \tag{39}$$

$$v_{i,ave} = \frac{v_i + v_{i+1}}{2}. \tag{40}$$

According to Equation (36), the sum of the set is equal to 1. If there are two nonzero variables, they should be adjacent and have a sum of 1. Here, $v_i'^2$ is a close approximation of $v_i^2$ and is used to directly present $v_i^2$ in the model. This explicitly defines a piecewise linear relationship between $v_i^2$ and $v_i$.

The maximum effort limited for the vehicle is $F_{max}$ during traction and $-F_{max}'$ during braking. The maximum power is $P_{max}$ during traction and $-P_{max}'$ during braking. $\eta_t$ and $\eta_b$ are the motor efficiency of the rail vehicle during traction and braking procedures. The model has the following constraints as in Equations (41) and (42).

$$-\eta_b F_{max}' \Delta d \leq E_i \leq \frac{1}{\eta_t} F_{max} \Delta d \tag{41}$$

$$-P_{max}' \frac{\Delta d}{v_{i,ave}} \eta_b \leq E_i \leq P_{max} \frac{\Delta d}{\eta_t v_{i,ave}}. \tag{42}$$

Because $1/v_{i,ave}$ and $v_{i,ave}^2$ are both non-linear term in the model, another set of SOS2 variables denoted by $\beta_i$ for $v_{i,ave}$ are used to present $1/v_{i,ave}$ and $v_{i,ave}^2$ by $v_{i,ave}$. Details are shown in Equations (43)–(47)

$$1 = \beta_{i,1} + \beta_{i,2} + \cdots + \beta_{i,j} + \cdots + \beta_{i,50} \tag{43}$$

$$0 \leq \beta_{i,j} \leq 1 \quad j = 1, 2, \cdots, 50 \tag{44}$$

$$v_{i,ave} = 50\beta_{i,1} + 49\beta_{i,2} + \cdots + (51 - j)\beta_{i,j} + \cdots + 1\beta_{i,50} \tag{45}$$

$$\frac{1}{v_{i,ave}'} = \frac{1}{50}\beta_{i,1} + \frac{1}{49}\beta_{i,2} + \cdots + \frac{1}{51 - j}\beta_{i,j} + \cdots + \frac{1}{1}\beta_{i,50} \tag{46}$$

$$v_{i,ave}'^2 = 50^2 \beta_{i,1} + 49^2 \beta_{i,2} + \cdots + (51 - j)^2 \beta_{i,j} + \cdots + 1^2 \beta_{i,50}. \tag{47}$$

There are other practical factors such as traveling time, gradient, and speed limit, which could influence the optimal trajectory. We note that $v_{i,ave}'^2$ is a close approximation of $v_{i,ave}^2$ and is used to directly present $v_{i,ave}^2$ where appropriate in the model. In the meantime, $\frac{1}{v_{i,ave}'}$ is a close approximation of $\frac{1}{v_{i,ave}}$ and is used to calculate relevant time information. If the maximum traveling time is $T_{max}$, the model imposes the constraint for the whole process, as shown in Equation (48).

$$T_{max} \geq \sum_{i=1}^{N-1} \Delta t_i = \sum_{i=1}^{N-1} \frac{\Delta d}{v_{i,ave}'}. \tag{48}$$

To obtain the optimal result in this MILP model with the consideration of motor efficiency, Equations (49)–(50) are imposed as constraints. Different from previous studies by [6,46], the efficiency of the motor is incorporated in this model. When the vehicle is braking, the electrical energy from the motor is assumed as a positive number. Alternatively, the electrical energy from the motor is a negative number when the vehicle is in traction. If $E_i \geq 0$, $\frac{E_i}{\eta_b}$ is larger than $E_i \eta_t$, Equation (49) will be applied while Equation (50) is relaxed. The calculated $E_i$ will be minimized while the efficiency $\eta_t$ is taken into account to represent the reduction rate when the electrical energy is transformed into

the kinetic energy, heat, and potential energy. If $E_i \leq 0$, $\frac{E_i}{\eta_b}$ is smaller than $E_i \eta_t$, Equation (50) will be applied while Equation (49) is relaxed. The kinetic energy will be transformed into electrical energy, potential energy, and heat. $\Delta h_i$ is the altitude change in each section.

$$E_i \eta_t - F_{i,drag}\Delta d - Mg\Delta h_i - \frac{1}{2}M(v_{i+1}'^2 - v_i'^2) \geq 0 \tag{49}$$

$$\frac{E_i}{\eta_b} - F_{i,drag}\Delta d - Mg\Delta h_i - \frac{1}{2}M(v_{i+1}'^2 - v_i'^2) \geq 0. \tag{50}$$

In order to ensure all of the braking and traction effort are constrained by the motor characteristics, Equations (51)–(54) are imposed. Equation (51) ensures the maximum kinetic energy reduction is no more than the one caused by the maximum braking effort. Equation (52) guarantees that the maximum kinetic energy increase is no more than the one caused by the maximum tractive effort. In such a way, the first two equations ensure the maximum tractive/braking efforts are not exceeded. Similarly, Equations (53) and (54) ensure the constraints of maximum tractive and braking powers are not violated.

$$-F_{i,drag}\Delta d - Mg\Delta h_i - \frac{1}{2}M(v_{i+1}'^2 - v_i'^2) \leq F_{b,max}\Delta d \tag{51}$$

$$F_{i,drag}\Delta d + Mg\Delta h_i + \frac{1}{2}M(v_{i+1}'^2 - v_i'^2) \leq F_{t,max}\Delta d \tag{52}$$

$$-F_{i,drag}\Delta d - Mg\Delta h_i - \frac{1}{2}M(v_{i+1}'^2 - v_i'^2) \leq P_{b,max}\Delta t_i \tag{53}$$

$$F_{i,drag}\Delta d + Mg\Delta h_i + \frac{1}{2}M(v_{i+1}'^2 - v_i'^2) \leq P_{t,max}\Delta t_i. \tag{54}$$

Based on the discussions in this section, we propose an MILP model, in which $v_0'^2, v_1'^2, \cdots, v_i'^2, \cdots, v_N'^2$ are the determining variables and the objective function is to minimize the total net energy consumption from the electrical motor as defined in Equation (33). We consider that no mechanical braking effort to enable a comparison with the method based on the PMP but other means of braking can be incorporated in the model by updating the constraints during braking in Equations (52) and Equation (54) using the maximum braking rate ([6,46]).

## 4. Results and Discussion

In this section, there are several different scenarios used to demonstrate the optimal speed trajectory under different constraints. The vehicle parameters are shown in Table 4. $v_0$ is the initial speed, and $v_t$ is the final speed. Without losing generality, $v_0$ is assumed to be larger than $v_t$. Four scenarios are used to demonstrate the optimal trajectory under different conditions. In these four scenarios, the traveling distance is assumed to be 18 km and the efficiency for traction and braking are assumed to be 60%, which could be updated for different traction systems. Similar efficiency data was proposed by [54]. The constant interval for the MILP method is selected to be 180 m, resulting in 100 intervals for the modeling.

**Table 4.** Modeling parameters for a typical urban rail vehicle.

| $M(\text{t})$ | $P_{max}(\text{kW})$ | $F_{max}$ or $F'_{max}(\text{kN})$ | $A_{max}(\text{m/s}^2)$ | $V_{max}(\text{m/s})$ | $A(\text{kN})$ | $B(\text{kN/m/s})$ | $C(\text{kN/(m/s)}^2)$ |
|---|---|---|---|---|---|---|---|
| 178 | 5000 | 200 | 1.2 | 45 | 3.6449 | 0.001710 | 0.01134 |

In the first three scenarios, we demonstrate the optimal partial speed trajectory on a flat track where only $V_{tc}$ is considered. $V_{tc}$ in the first scenario is based on the condition of $V_{tc} > v_0$, $V_{tc}$ in the second scenario is based on the condition of $v_0 > V_{tc} > v_t$, and the third scenario is based on the condition of $V_{tc} < v_t$. In addition, taking both $V_{tc}$ and $V_{bc}$ into consideration, in Scenario 4, we assume there are two different slope rates, which are uphill and downhill in the entire trajectory.

In the following sections, Figures 11–14 demonstrate the results for these four cases using MILP; Figures 15–18 show the results based on the PMP. Each scenario is selected based on different set average speed based on the set total journey time and journey length. The final results in all MILP cases include the optimal speed trajectory, efforts applied, net energy, the total time in theory and the actual total time. The net energy indicates the electrical energy consumption including traction energy and regenerative energy. The total time in theory is the total travel time calculated by the MILP model. The actual total time is the traveling time calculated using a post-process calculation after the trajectory has been obtained using small distance step iteration. The small difference between these two time values well reflect the accuracy of the currently used MILP model. Similar to the results obtained by MILP, the ones using PMP will additionally provide the evolution curves of co-state variable $\theta$.

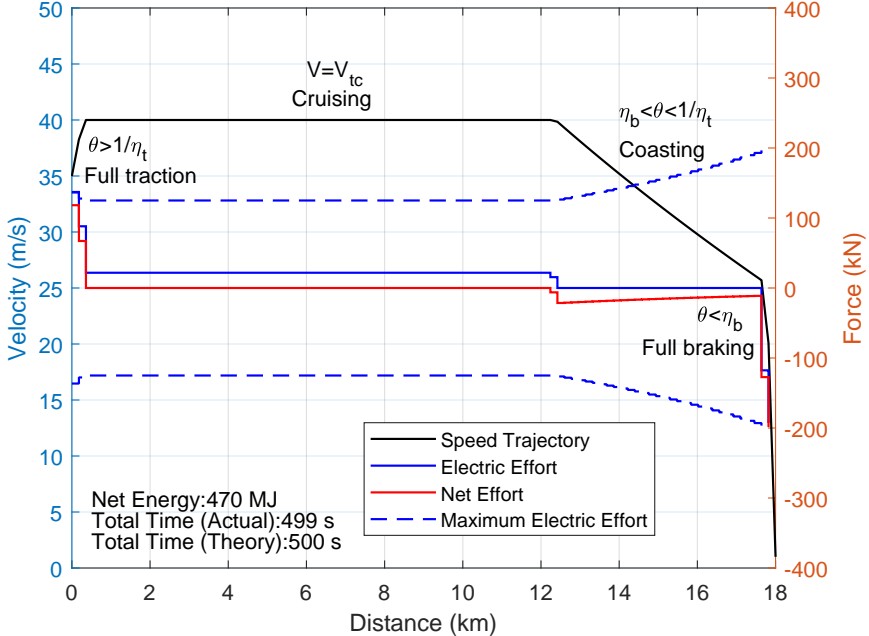

**Figure 11.** Optimal speed trajectory for Scenario 1 based on MILP.

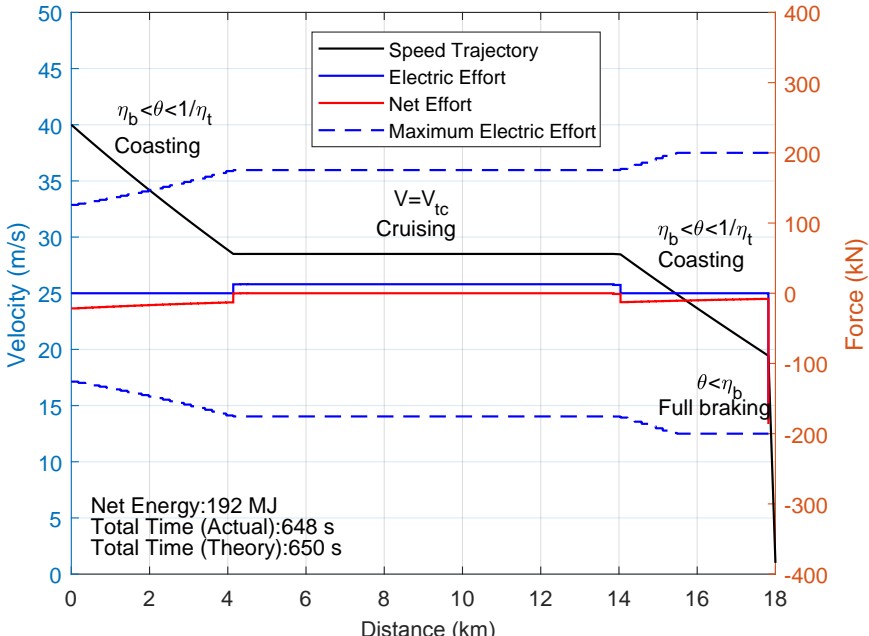

**Figure 12.** Optimal speed trajectory for Scenario 2 based on MILP.

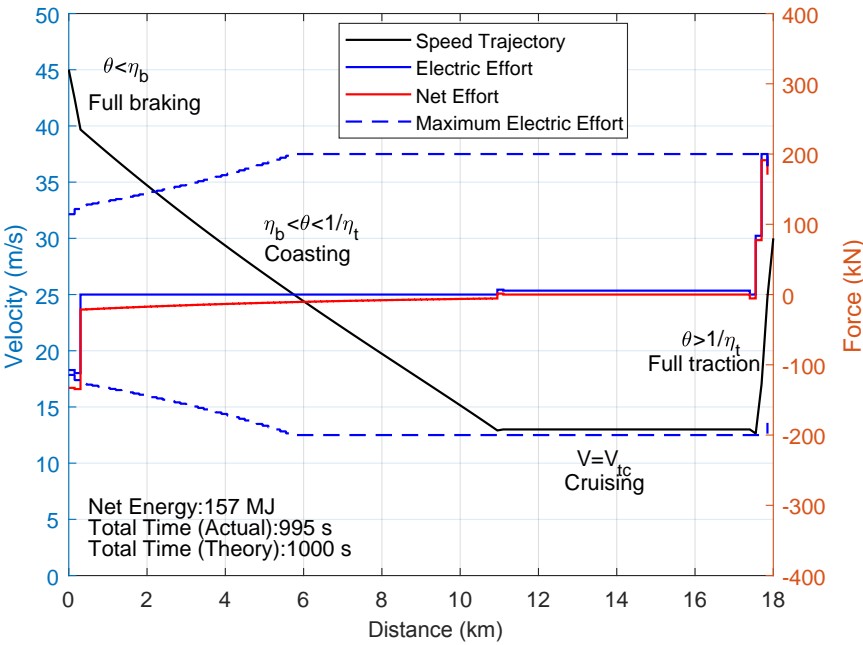

**Figure 13.** Optimal speed trajectory for Scenario 3 based on MILP.

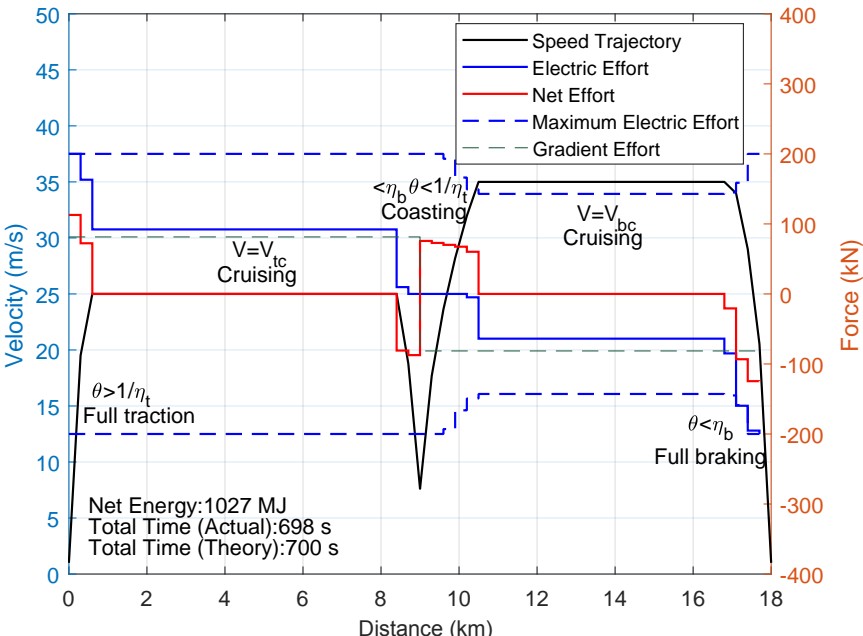

**Figure 14.** Optimal speed trajectory for Scenario 4 based on MILP.

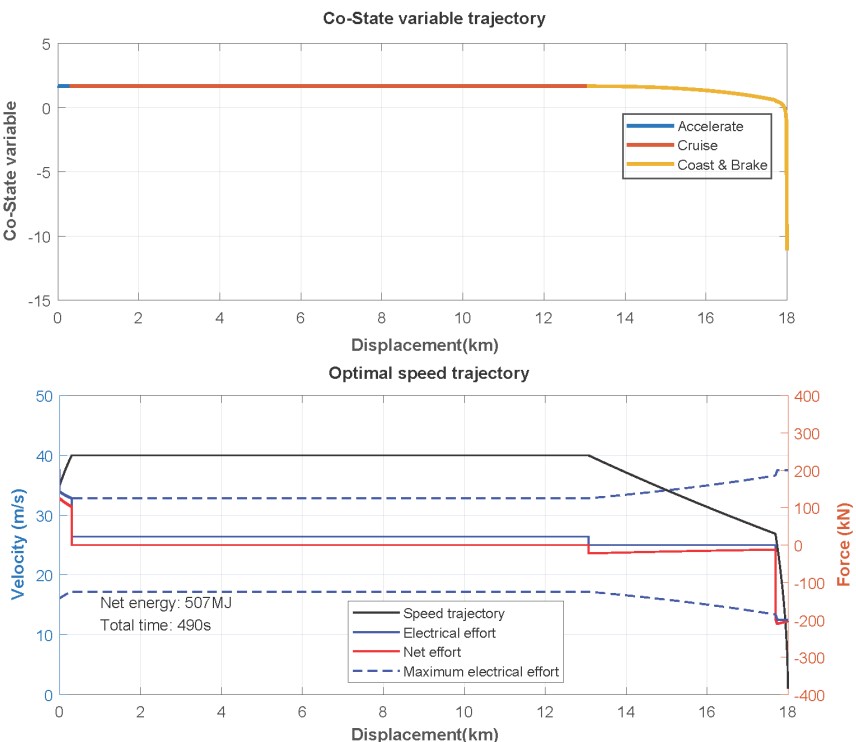

**Figure 15.** Optimal speed trajectory for Scenario 1 based on the PMP.

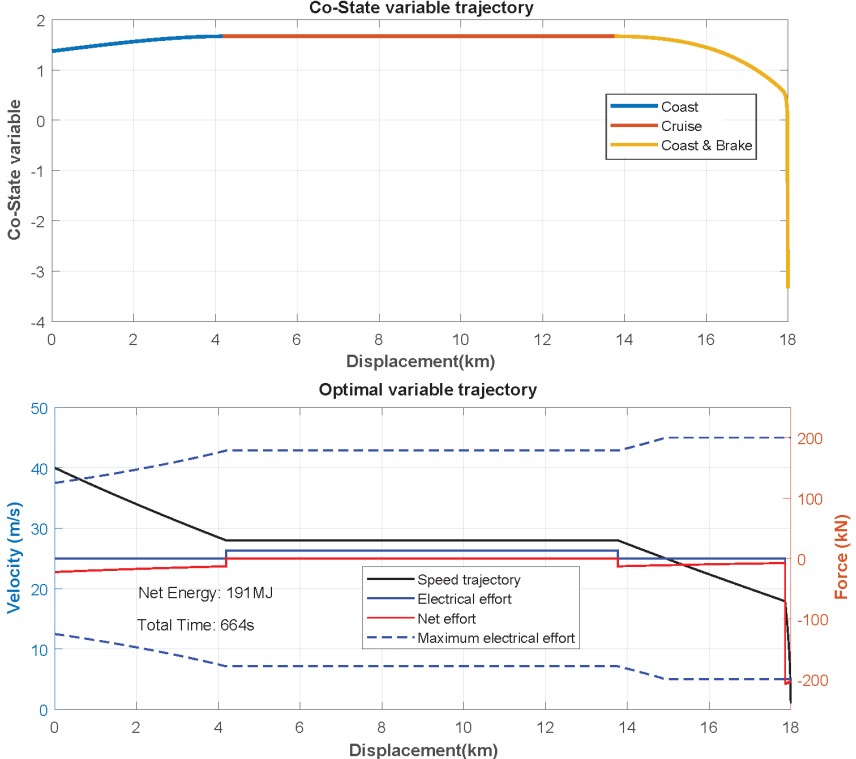

**Figure 16.** Optimal speed trajectory for Scenario 2 based on the PMP.

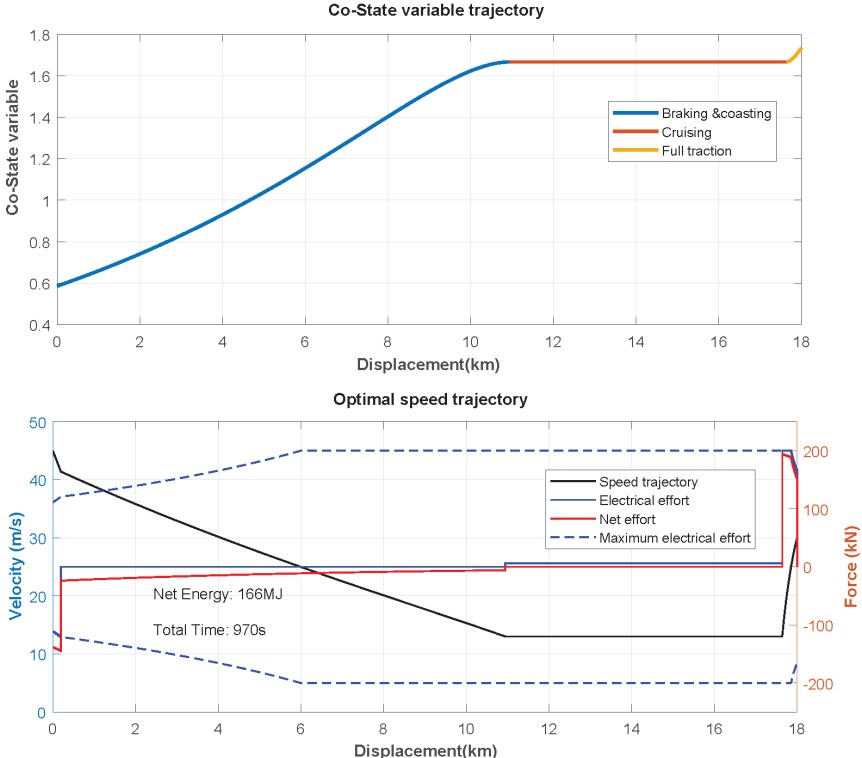

**Figure 17.** Optimal speed trajectory for Scenario 3 based on the PMP.

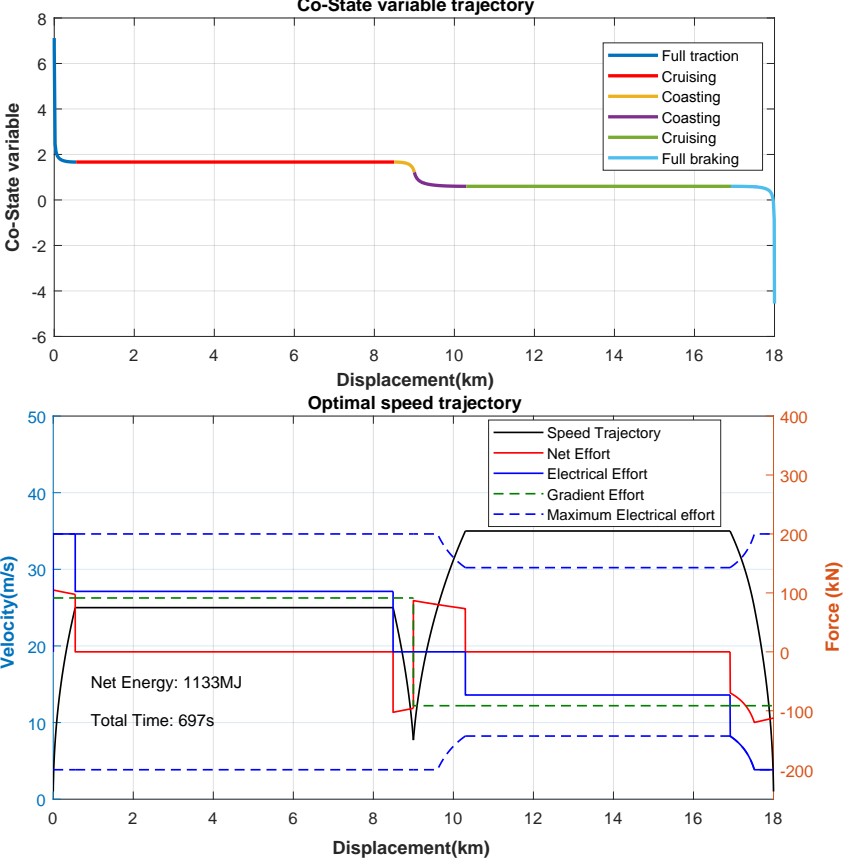

**Figure 18.** Optimal speed trajectory for Scenario 4 based on the PMP.

### 4.1. Scenario 1: With a Set Average Speed of 36 m/s, $V_{tc} > v_0$, $v_0$ = 35 m/s, $v_t$ = 1 m/s

Figures 11 and 15 demonstrate the optimal speed trajectory using methods based on MILP and the PMP respectively for a scenario with an initial speed of 35 m/s, a final speed of 1 m/s, and a trip time of 500 s. In Figure 11, according to the applied electrical effort, there are four operations existing in this trajectory, i.e., traction, cruising, coasting, and full braking operations. As shown in Figure 11, we can easily spot the cruising speed at 40 m/s, and this can be regarded as $V_{tc}$. The initial state of the vehicle could be considered as $v < V_{tc}$ and $\theta > \frac{1}{\eta_t}$, which corresponds to the deduction of Case 9.2, where the vehicle will perform the full traction with the increase in speed and decrease of $\theta$ until $v = V_{tc}$ and $\theta = \frac{1}{\eta_t}$, and takes the cruising operation as the subsequent operation in the optimal speed trajectory. Because the final speed $v_t < V_{tc}$, the vehicle will perform the operations in Case 8, which includes the coasting and full braking. $\theta$ will keep decreasing until it goes below $\eta_b$, and the optimal operation of train can be analyzed as in Case 7, in which the train will conduct full braking toward the speed $v_t$. Similar observations can be made for the results from the PMP as shown in Figure 15 with four operations coexisting in the trajectory. In Figure 15, we are able to know the detailed trajectory of the co-state variable. The main challenge for the method based on the PMP is that the algorithm needs to search for the corresponding $V_{tc}$ to obtain the final trajectory, while the one based on MILP needs not. Another interesting observation is that results obtained from the method based on the PMP have continuous characteristics, while the ones obtained from the method based on MILP shows discrete characteristics.

### 4.2. Scenario 2: With a Set Average Speed of 27.7 m/s, $v_t < V_{tc} < v_0$, $v_0$ = 40 m/s, $v_t$ = 1 m/s

Figures 12 and 16 demonstrate the optimal speed trajectory using methods based on MILP and the PMP respectively in Scenario 2 with an initial speed of 40 m/s, a final speed of 1 m/s, and a trip time of 650 s. In Figure 12, according to the applied electrical effort, there are four operations performed in the speed trajectory which are coasting, cruising, coasting, and full braking. Therefore, the cruising speed of the optimal speed trajectory is 29 m/s in this scenario, which can be regarded as $V_{tc}$. During the first and second operations, which are from around 0 to 4200 m and from 4200 to 13,800 m, the vehicle conducts the operations as reflected by the deduction of Case 5 with $v > V_{tc}$ and $\eta_b < \theta < \frac{1}{\eta_t}$. After cruising, the vehicle will coast and decrease the speed, which is in line with the deduction in Case 8. Since the final speed $v_t$ is less than $V_{tc}$ and due to the continuity of $\theta$, the deduction of Case 7 will be applicable for the optimal operation of train, and the vehicle will conduct full braking operation to the final speed $v_t$.

### 4.3. Scenario 3: With a Set Average Train Speed of 18.0 m/s, $V_{tc} < v_t$, $v_0$ = 45 m/s, $v_t$ = 30 m/s

Figures 13 and 17 demonstrate the optimal speed trajectory using methods based on MILP and the PMP respectively for Scenario 3 with an initial speed of 45 m/s, a final speed of 30 m/s, and a traveling time of 1000 s. According to the applied electrical effort, there are four operations which are full braking, coasting, cruising, and full traction. Since the traveling time is relatively long, $V_{tc}$ is lower than the initial and final speed. With this traveling time for the whole trajectory, optimal results show that the cruising speed is 13 m/s, which could be taken as $V_{tc}$. At the beginning of the trajectory, $v > V_{tc}$, and the vehicle is performing the full braking operation. Therefore, $\theta > \frac{1}{\eta_t}$ and $\frac{d\theta}{dx} < 0$ could be obtained according to the deduction of Case 4. With the decreasing of $\theta$, the operation of the vehicle will turn to coasting and than cruise as the deduction of Case 5.1 with $\theta$ coming to $\eta_b < \theta < \frac{1}{\eta_t}$ and $\frac{d\theta}{dx} > 0$, which matches the trajectory between around 500 to 16,500 m in the optimal speed trajectory from MILP.

After the cruising operation, the vehicle takes a short coasting between 16,500 and 17,500 m before it conducts the full traction operation to reach the final speed. This short coasting operation is considered to be mainly due to the discretization of search space and operation constraints of the model.

Similar observation can be made on the braking effort imposed at the beginning of the journey. During the full traction operation when $v > V_{tc}$, $\theta > \frac{1}{\eta_t}$ could be obtained from the deduction of Case 6.

### 4.4. Scenario 4: With Uphill and Downhill Slopes, $v_0$ = 1 m/s, $v_t$ = 1 m/s

Figures 14 and 18 demonstrate the optimization result using methods based on MILP and the PMP respectively from Scenario 4, which contains an uphill slope between 0 and 9000 m and an downhill slope between 9001 and 18,000 m. The forces imposed on the vehicle due to gravities have been shown in the figures. Based on the MILP optimization result, the speed of cruising operation under partial traction and partial braking effort are applied, which are 25 and 35 m/s, respectively. We regard these two values as $V_{tc}$ and $V_{bc}$ during the uphill and downhill sections in the optimal speed trajectory. The cruising speed in this scenario should satisfy $v_t < V_{tc} < v_0$ and $v_t < V_{bc} < v_0$. For the beginning of the uphill section, the vehicle will perform the full traction operation and then cruising. $v \leq V_{tc}$ will be satisfied. According to the deduction of Case 9.2, $\theta > \frac{1}{\eta_t}$ and $\frac{d\theta}{dx} < 0$ should be satisfied at the same time. The operation will turn to coasting as the deduction of Case 8 with the decreasing of $\theta$, which is in accordance with the coasting operation in optimal speed trajectory from around 8500 to 9000 m. At the beginning of the downhill section, the vehicle performs the coasting operation and $v < V_{tc}$ which matches the deduction of Case 5.1. According to Case 5.1, the vehicle will perform coasting operation and then cruising with $\eta_b < \theta < \frac{1}{\eta_t}$ and $\frac{d\theta}{dx} < 0$. The deduction of Case 5.1 could also match the operation from optimal speed trajectory between 9000 and 17,000 m. The full braking operation will be performed toward the end of trajectory, which could be matched with the deduction in Case 4. It could also be observed that the motor of the vehicle is working under different modes when the vehicle is doing cruising operation during uphill and downhill. As is shown in Figure 14, the motor outputs braking effort when the vehicle is going downhill and outputs traction effort when it is going uphill. Similarly, the result shown in Figure 18 provides a verification on the ones obtained from MILP, and it is with smoother curves due to the modeling characteristics of PMP.

### 4.5. A Summary of Optimization Results

We will now present Table 5 to summarize the optimization results for the four cases. For each case, we show the total energy for optimized speed trajectory using both methods, the computational time, and the journey time for both methods. It can be observed that in general the algorithms based on the PMP have a longer computational time compared to the MILP algorithm.

**Table 5.** Optimization results for the four cases.

|  | Scenario 1 | | | Scenario 2 | | | Scenario 3 | | | Scenario 4 | | |
|---|---|---|---|---|---|---|---|---|---|---|---|---|
|  | E(MJ) | $T_1$(s) | $T_2$(s) | E(MJ) | $T_1$(s) | $T_2$(s) | E(MJ) | $T_1$(s) | $T_2$(s) | E(MJ) | $T_1$(s) | $T_2$(s) |
| PMP | 507 | 490 | 40.75 | 191 | 664 | 52.03 | 166 | 970 | 56.16 | 1133 | 664 | 80.68 |
| MILP | 519 | 499 | 4.30 | 192 | 648 | 1.02 | 157 | 995 | 3.25 | 1027 | 698 | 0.66 |

Note: E: Energy consumption $T_1$: Journey time $T_2$: Computation time.

### 4.6. A Critical Analysis of the Results from Both Methods

It is observed that many underlying factors can potentially lead to the difference between the optimization results obtained from MILP and the PMP. The difference of the results includes different operation switch locations, different traction and braking efforts during train operation, and different total journey time and energy consumptions. For example, in Scenario 1, it can be seen from Figures 11 and 15 that the location for the start of coasting is substantially different: around 12.3 km for MILP and around 13.1 km for the PMP. These underlying factors leading to this phenomenon are not normally available for direct observations as it is related to the algorithm solver and the fundamental algorithm design, but can be indirectly reflected by the final results.

We may argue that the difference is partly caused by the different algorithm design of both methods, i.e., one indirect and the other direct, in optimal control. In the PMP method, we need to first determine the constant cruising speed and then obtain the entire trajectory using the search methods as proposed in the paper. After we evaluate the journey time, we adjust the constant cruising speed to update the solution. This is a key reason that it is not as easy as it is for the MILP method to achieve an exact journey time. On the other hand, the results obtained from MILP is relatively straightforward. We include a time constraint in the model, and the result will automatically provide the final optimal solution obtained by the solver. In summary, a different algorithm design may unavoidably lead to different optimization outputs.

Another important underling factor leading to result discrepancy is the modeling discretization and precision of the MILP model. In the MILP method, we assume that the train is running with a constant tractive or braking effort within one distance interval. Due to the distance discretization, the algorithm sometimes is not able to obtain the maximum braking effort in a short distance. Take Figure 13 as an example. Based on the PMP optimality analysis as shown in Case 4 in Table 3, the initial co-state variable leads to a full braking and the value of co-state variable will be further reduced due to a negative derivative. Such a reduction subsequently gives rise to coasting and cruising operations afterward. As a result, at the beginning, the train is supposed to conduct a full braking operation. However, the actual braking effort obtained by the model is the average braking effort, not the changing maximum braking effort. With a reduction of $\Delta d$, this situation can be improved as average efforts will be closer to the maximum braking efforts. Similarly in the same case, during the full acceleration at the end of the journey, we are able to demonstrate stair-type tractive efforts reaching a maximum effort of around 200 kN.

## 5. Real-World Case Study—Dutch Railway Corridor

Here, we use the the real-world railway system to validate our MILP model. The case study is based on the real schedule and field data of Dutch railway corridor from Utrecht (Ut) to Houten (Htn), where the Utrecht Lunetten (Utl) is the intermediate station, see Figure 19. The field data for rolling stock and the running schedule are shown in Tables 6 and 7.

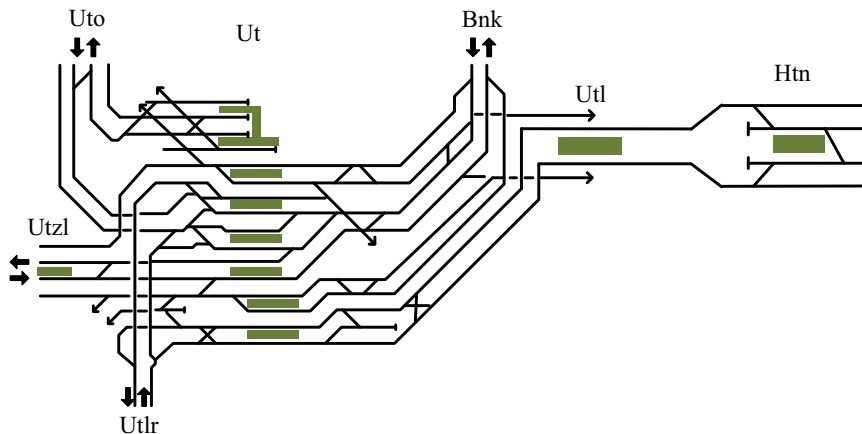

**Figure 19.** The railway track between Utrecht and Houten.

**Table 6.** Parameters for a dutch railway vehicle.

| $M(\text{t})$ | $P_{max}(\text{kW})$ | $F_{max}(\text{kN})$ | $A_{max}(\text{m/s}^2)$ | $\Delta d(\text{m})$ | $A(\text{kN})$ | $B(\text{kN/m/s})$ | $C(\text{kN/(m/s)}^2)$ |
|---|---|---|---|---|---|---|---|
| 233.2 | 1918 | 170 | 0.8 | 300 | 2.2601 | 0.00125 | 0.00006 |

**Table 7.** Track length and running time from Utrecht (Ut) to Houten (Ht).

|         | Track Length (m) | Scheduled Running Time (s) |
|---------|------------------|----------------------------|
| Ut–Utl  | 4330             | 240                        |
| Utl–Htn | 3619             | 180                        |

The maximum mechanical power and traction force are given and the maximum braking rate is the only accessible parameters to measure the train acceleration and deceleration motions. Due to the flexibility of our proposed model, the values for each distance interval $\Delta d_i$ are not strictly the same and can be changed to fit the different route length, the borders of speed limit switch points, and the gradient switch points. Since we only consider the minimization of the mechanical energy in this case and there is no regenerative braking in this railway corridor, the motor efficiency $\eta_t$ is set to be 1, and $\eta_b$ 0.

The speed limit and the route gradient of this corridor are shown in Figure 20. The same corridor is studied in [4] using the pseudospectral method (PSOPT), and the speed trajectories of it is also put together with the results of our model. In Figures 21 and 22, we show the optimal speed trajectories and the corresponding traction/braking curves yielded by our MILP method and the PSOPT method. The two speed profiles are similar and the MILP model brings about a more smooth trajectory, while PSOPT brings about more fluctuating ones. The traction and braking curves for two methods remain similar.

The results including the energy consumption and the computation time are shown in Table 8. The energy consumption and the computation time of this railway corridor obtained by the two methods are not significantly different. The comparison shows that the proposed MILP method is flexible and robust in dealing with the real-world case even with complex route constraints.

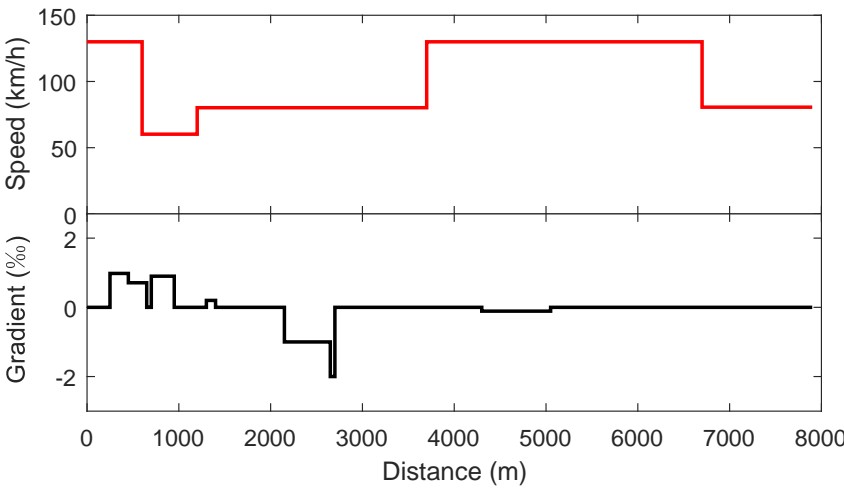

**Figure 20.** Speed limit and gradient information.

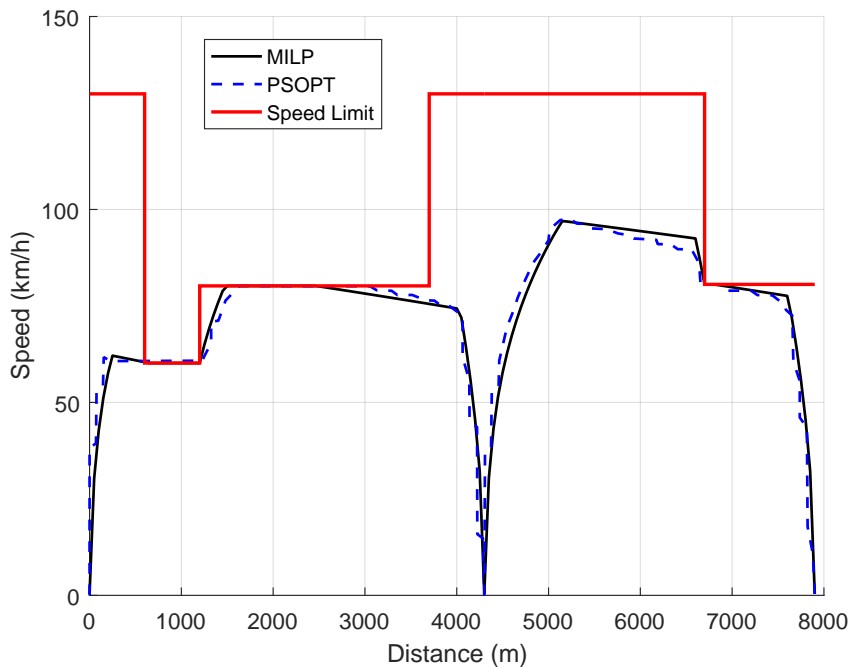

**Figure 21.** Optimal speed trajectory for real case based on MILP and the pseudospectral method (PSOPT).

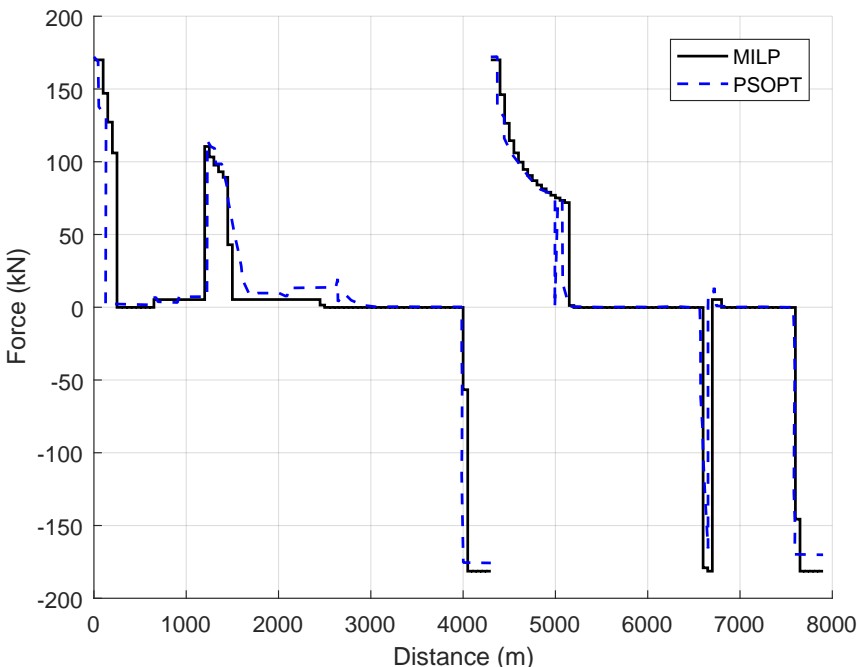

**Figure 22.** Optimal traction/braking curves for real case based on MILP and PSOPT.

**Table 8.** Results of the train speed trajectories optimization.

|          | Energy Consumption (MJ) | | Computation Time (s) | |
| -------- | ----- | ----- | ----- | ----- |
|          | **MILP** | **PSOPT** | **MILP** | **PSOPT** |
| Ut–Utl   | 70.86 | 70.92 | 7.55 | 3.80 |
| Utl–Htn  | 87.95 | 85.82 | 2.00 | 2.20 |

## 6. Conclusions and Future Work

This paper is focused on an adaptive partial speed trajectory optimization problem with considerations of motor efficiency for both traction and regenerative braking. Different from previous studies, the adaptive partial speed trajectory optimization is targeted on cases with non-zero initial and final speed states and has completely removed the assumption of speed monotonicity as proposed in previous papers [6,47], leading to its flexibility on various engineering cases. In particular, motor efficiency for both traction and regenerative braking as a key influencing factor for energy consumptions has been considered in this paper. With explicit discussions on the optimality analysis for the optimal train operation between two non-zero speed points, two methods based on the PMP and MILP are applied to obtain the optimal partial speed trajectory respectively.

We draw the following conclusions.

- The PMP is used to provide a theoretical analysis, and a numerical algorithm can be developed to obtain the optimal speed trajectory. In the meantime, the PMP could be used to derive theoretical deductions for different situations and verify the results obtained from mathematical programming, as has been done in this paper.
- Mathematical programming such as MILP provides a handy tool to optimize the train speed trajectory. This method is based on the mathematical programming method and can be easily modified and improved to incorporate other engineering constraints. For example, based on the proposed MILP model, we can take into account the route information, motor efficiency, and on-board energy storage devices as shown in the real case study, compared and contrasted in another methods proposed in other papers [4].
- Discretization introduced in mathematical programming will inevitably introduce modeling errors and reduce the result precisions, though the requirement for speed control precision in large inertial systems such as rail transportation is usually not very high. On the other hand, the search space for the model based on mathematical programming will largely affect the calculation results and demand more delicate design of the model, sometimes with a sacrifice on the modeling precision.

In the theoretical analysis, we make a strong assumption with no gradients and speed limits, which limits the generality of our discussions. The detailed case studies on different scenarios demonstrate the robustness of the proposed methods based on the PMP and MILP, but further studies are still needed by implementing more constraints to reflect more complicated engineering operations. The discretized modeling in the MILP-based method takes the average tractive/braking effort within a discretized distance and is unable to reach the changing maximum effort during acceleration. This will lead to sub-optimal solutions. A detailed critical discussion on the results from both methods can be found in Section 4.6.

Future work needs to address issues arising from emerging traction technologies such as on-board energy storage devices, three-dimensional motor efficiency maps, and multiple train energy interactions within an electrical network. The proposed MILP model can be well extended for more complex integrated optimization by considering multi-train interactions and real-time signaling constraints.

**Author Contributions:** Z.T. and S.L. conceived and designed the experiments; Z.T., K.B., S.Z., and C.W. performed the experiments; Z.T., S.L., and C.W. analyzed the data; J.Y. and F.X. contributed to the paper structure and framework design; Z.T., K.B., and S.L. wrote the paper.

**Funding:** This project was supported in part by the 2016 NSFC Young Scientist Program Project No. 61603306, in part by the collaborative project with SDIC Baiyin Wind Power Co. (Contract No. 2016-GTJCT-JF-29), in part by the 2017 NSFC Program Regional Collaboration and Exchange Project No. 61763016, in part by the Research Development Fund RDF-14-01-15 and RDF-16-01-42 at Xi'an Jiaotong-Liverpool University, in part by the National Key R&D Program of China No. 2017YFB1201105-12, in part by the Suzhou Science and Technology Program under No. SZS201613, in part by the 2017 Summer Undergraduate Research Fellowship (SURF) program at Xi'an Jiaotong-Liverpool University (Project ID: 201731).

**Acknowledgments:** The authors would like to express their sincere gratitude to all our respective reviewers for their pertinent and constructive comments.

**Conflicts of Interest:** The authors declare no conflict of interest.

## Abbreviations

The following abbreviations are used in this manuscript:

| | |
|---|---|
| MDPI | Multidisciplinary Digital Publishing Institute |
| PMP | Pontryagin's maximum principle |
| ATO | automatic train operation |
| MILP | mixed-integer linear programming |
| ACO | ant colony algorithm |
| GA | genetic algorithm |
| SQP | sequential quadratic programming |
| PSOPT | pseudospectral method |

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
