# Peer review of "Adaptive Partial Train Speed Trajectory Optimization"

_energies, doi:10.3390/en11123302_

Round 1

Reviewer 1 Report

Dear Authors, 

your work deals with a partial speed trajectory optimization problem for train operations between two speed points with the given traveling time and distance.

In the following there are few minor comments:

- please define the meaning of "PMP" and "MILP" in the introduction;

- page 2 line 74: please insert the number of the section (it should be section 1);

- table 1 is not recalled in the main text;

- page 4: please correct "psudospectral"

- please add a section with the list of acronyms\symbols used;

- the lowercase greek letter phi is used with reference to 2 different quantities, please check the text;

- page 16: the section "Algorithm 1" is not readable;

- page 17: the section "Algorithm 2" is not readable;

- page 19: please check the sentence "v'i,ave2 is a close approximation of v'i,ave2";

Thank you

Best regards

Author Response

Please refer to our PDF file for detailed responses to our reviewer's comments. 

Reviewer 2 Report

Recently, it has been developed as an autonomous driving system through unmanned driving. Also, researches are being actively carried out to obtain the optimal operating efficiency by computerized driving information by unmanned operation.

This paper presents the problems in the automatic operation of the railway car and suggests the control logic to solve them.

We suggest to modify the following parts

The meaning of the title of the paper is unclear. It should be modified to be able to represent the important research content of the  paper.

The ATO speed profile, which is considered in this study, should be clarified, and the train should be controlled based on it. Identify the current control problem based on the content presented.

   3. I do not understand exactly what the situation described in each case is. Please draw a picture in more       

       detail.

You divided the speed of the train into three parts. What is the basis for it? Is it a high-speed train, a low-speed train, or an urban train? Or did you split the speed of a particular train?

Reliability of the model is very important for verification of the control logic. In this paper, however, there is a lack of modeling.

       How did you model the train using table 4 data? Please present the modeling expression or model.

Verification of the control logic is appropriate for scenarios that might occur in real rail vehicles. What is the basis for the verification scenario of this paper?

It seems that the experiment is applied to the actual vehicle. Please present the comparison data before and after applying the control logic presented in this paper, and explain the improvement point.

Author Response

Please refer to PDF file for detailed responses to reviewers' comments. 

Reviewer 3 Report

This paper proposes and compares two methods for the optimisation of speed profiles to minimise energy consumption. Overall, the paper is a good candidate for this journal, but I have three principal concerns. Firstly, despite a key contribution of the paper being to propose methods that include traction equipment efficiency, efficiency appears to be assumed constant, which is a significant simplification. Secondly, both the validation and critical analysis of the proposed methods are not comprehensive enough. Finally, I was not able to independently replicate the speed profile results, suggesting there may be errors in either the paper’s description or the method itself. As such, some further work is necessary before I can recommend acceptance. I believe the amount of additional research required is small by comparison with what has already been done, and would significantly improve the case for publication. Likewise, a small amount of additional analysis would significantly improve the paper. More detailed feedback is provided below:

1. Consideration of energy efficiency in the calculation of optimal speed profiles is stated as a key contribution of this paper (line 70). It was demonstrated by research 20-30 years ago (based on real-world measurements) that variable efficiency across the speed and power range can have a significant effect on the optimal control strategy (see Van Dongen and Schuit/Franke et al., both cited in your reference [49]). Many previous theory-based papers (using PMP) ignore efficiency and also regenerative braking, and it would be useful to analyse the implications from previous research for your work in more detail in the literature review, rather than simply citing a large number of papers and providing only a brief description of their content.

2. It appears from line 381 (and elsewhere) that you have treated the efficiencies as constant, rather than defining eta_t as a function of both v and u_t (and likewise for eta_b for v and u_b). However, as per point [1.], variable part-load efficiencies across the speed range can have a significant effect on the optimum control strategy. As such, your work should prove that the 3D efficiency map alluded to in line 528 can successfully be incorporated into your proposed method, rather than mentioning it only as future work. Otherwise, you have not eliminated the possibility that variable efficiency is an example of a practical constraint that means ‘speed trajectory optimisation becomes an NP-complete problem’ (line 146). The potential usefulness of your proposed method is limited if it cannot improve on existing practice when applied to real world problems, undermining the potential contribution of your work. The optimisation of speed profiles to reduce energy consumption has been the subject of a large number of papers in recent years, and so a significant contribution to knowledge is necessary to justify publication of more research in this area.

3. A term for train mass appears to be missing from Equation (1b). Alternatively, are the terms on the right hand side of the equation defined as accelerations rather than forces?

4. Some of the notation is used before it is introduced or defined fully. The clarity of the paper would be improved if these are either: a) fully defined when first used, or b) a separate list of notation provided.

5. In the second line of the paragraph between Equations (48) and (49) on page 19 (line numbers are missing from this section in my review copy of the manuscript), v’_i,ave^2 appears twice, should one of these be v_i,ave^2?

6. I cannot replicate your speed profile results in Section 4 – specifically the deceleration when coasting. For example, using the data given in Table 4 and looking at Case 2 where the train immediately coasts from an initial speed of 40 m/s, my calculations suggest that the train speed after 4 km should be around 33 m/s, not the 28 m/s in your results., Please check whether this is due to an error in Table 4, an error in the input data used in your calculations, or indicative of a problem with the method itself. My results were obtained by using a spreadsheet to calculate the total resultant force (hence change in speed) and the cumulative distance travelled at 1 second intervals, to build up the speed/distance profile. As an example, the initial force at 40 m/s is 2.0895 + (0.0098 x 40) + (0.0065 x 40 x 40) = 12.9 kN (corresponding to an acceleration of -0.072 m/s/s for a 178 t train, ignoring rotational inertia), but Figures 13 and 14 appear to show a rather larger force of approximately 20 kN.

7. I am not convinced by the brief statement on line 415 that the continuous or discrete characteristics of PMP or MILP respectively are responsible for the (significant) difference in energy consumption. For case 1, it can be seen in Figures 11 and 12 that the location for the start of coasting is substantially different: around 12.3 km for MILP and around 13.1 km for PMP. This also has a corresponding effect on the brake entry point, and overall is likely to have a significant effect on the energy consumption. Furthermore, the total running time is different. Such discrepancies also feature in other scenarios in Table 5, and are large enough that the paper should investigate the underlying reasons in more detail rather than dismiss them as irrelevant.

8. The cases in Section 4 are relatively simple scenarios. As such, it would be practical to carry out a comprehensive evaluation of the energy consumption of a complete set of viable speed profiles (for example, for case 1: evaluate total energy consumption for v_cruise = 35, 36…45 and s_coast/(s_coast+s_brake) = 0, 0.1…1, subject to the overall constraint of T = 500) using single train simulation, or relatively simple numerical methods such as those mentioned for point [6.]. This would objectively show whether the proposed methods are indeed finding the minimum energy consumption, and it therefore represents a more comprehensive validation of their effectiveness. At present, the two methods are compared against each other, but there is no indication of whether the two profiles are close to or at the actual minimum energy consumption. As noted in point [7.], there are also significant differences between the results from each method, which further undermines the paper’s claims of a full validation (line 508). Furthermore, this exercise could also be used to investigate the trade-off between precision and search space size identified on line 525.

9. Applying your PMP method to the real-world case study in Section 5 to provide an additional comparison would also improve the paper.

10. The placement of the figures and tables in Sections 4 and 5 could be improved to make the paper easier to read, as sections that relate to each other have been split across multiple pages. For example, Section 4.2, Figure 13 and Figure 14 would benefit from being directly adjacent to each other.

11. The results for scenario 3 include some braking early in the journey followed by acceleration later in the journey, which is an unexpected result for an optimal speed profile where efficiency is included and there is no line speed limit constraint. Furthermore, the resulting force from full braking in Figure 15 (around 50 kN) is significantly different to the force from full braking in Figure 16 (around 110 kN). ‘Full’ braking implies to me that u_b = 1, suggesting that the force should be the same for both methods – is this the intended meaning? The underlying reasons for both of these observations should be investigated as part of the discussion of the results.

12. The discussion of the results and conclusions are rather superficial, often limited to single statements like line 508 that claim there is a very close match between the methods, despite the incomplete validation and examples (such as in point [7.]) where the match between the two methods is not close and further analysis of the discrepancies is warranted. Likewise, potential weaknesses of the methods and the implications of assumptions inherent in the methods are only examined briefly.

Author Response

(The authors gave the same response as above.)

Round 2

Reviewer 1 Report

Dear Authors, 

thank you for the revised version of your manuscript.

Best regards

Author Response

Dear Reviewer 1,

We are grateful for your valuable review comments to further improve our paper's quality.  

Kind regards, 

Shaofeng Lu on behalf of all co-authors 

Reviewer 2 Report

 ATO is not a simple safety device. It is a controller for automatic operation of trains. Therefore, the control algorithm you presented is operated on this controller. I am wondering if you are aware of the above, and it is very important for the person studying the control algorithm to know where and how this control logic is used.

It is still unclear what exactly the algorithm you are presenting is used for. Does presenting a control algorithm mean that a train is automatically operated? If so, does the train recognize the basic mechanism of automatic operation? Add an explanation of where your proposed control logic is used in the correct part.

Existing train systems are expert systems that rely on expert experience or long operating experience. In order to further evolve this, we are carrying out research that is applying gull control technology. Therefore, it is important to clarify the following points. The control logic you used did not clearly divide the speed of the train. Why did you split the simulation scenario into speeds? And give a clear figure of the speed. The expressions of high speed and low speed are too unclear.

Author Response

Dear Reviewer, 

We have included our detailed responses to your valuable comments in the attached PDF file. We appreciate your time and efforts which have make our paper quality much improved. 

Kind regards, 

Shaofeng

Reviewer 3 Report

Thank you for the detailed and useful feedback. In general, you have written some very good explanations in your response to the first review, but these are not always reflected in the paper itself. Including some or all of these explanations in the paper (rather than only in the response to reviewers) would improve it further. There are also a few other minor points remaining. The details are given below, using the relevant numbered sections from the first review:

1. The additional text you have added satisfactorily addresses this point. However, ‘electrical energy from the power network’ is too ambiguous; I suggest ‘electrical energy of individual trains’ (or something similar). This makes it clear that the scope of the paper is optimisation of speed profiles for individual trains only, and that it does not cover network receptivity and transmission losses under the influence of multiple trains running within a network.

2. The first paragraph of your response to the review comments is much better than the single sentence you have added to the paper on line 32. Including more of this detail in the paper itself (to provide greater/more objective justification for the choices you made) would improve it further.

Furthermore, I suggest also including a sentence that explicitly confirms that the method is capable of using a variable efficiency function, although treated as a constant for this paper. This could perhaps be placed on either line 34 or line 530.

3. -

4. -

5. Thank you – don’t worry about the missing line numbers, this is only a minor formatting issue with the review copy of the manuscript, and had no influence on my review of the paper’s actual content.

6. Thank you for the corrections, the updated values agree with my calculations (I assume you meant Table 4 rather than Table 6?).

7. Your response to this point is a good example of the more detailed critical analysis of the method and results that is still missing from the paper itself – adding this (perhaps as a new Section 4.6) would significantly improve it. You have presented a good description of the results, but currently there is little actual discussion of them. The paragraph you added to Section 4.3 is a start, and could perhaps be moved to this new discussion section, as discretisation applies to all scenarios.

I do agree that the results from both methods are broadly similar and fulfil your aim of showing they are comparable, but I cannot support the strength of your assertions that ‘both methods…offer very similar solutions’ and ‘a high agreement to each other has been observed’ when the results show differences of the order of 10% in energy consumption (scenario 4) or differences of nearly 1 km for the coasting point (scenario 1). Furthermore, you state that there is ‘an accepted range of journey time difference’, but do not justify why the observed range is acceptable, what size difference would be considered unacceptable, or whether the differences in journey time could significantly affect the comparison of energy consumption. Toning down the statements about the strength of the agreement, and providing some justification for why the differences are small enough to accept, would mitigate potential criticism on this point and complement the additional discussion suggested in the previous paragraph.

8. -

9. Two minor points on line 510 I missed in the original review: if eta_t = 1, it would be better to replace ‘traction energy’ with ‘mechanical energy’ to avoid ambiguity. Furthermore, I presume eta_b = 0 as there is no regenerative braking – it would be worth stating this explicitly.

10. -

11. To avoid any potential misunderstanding here: my original concern was not that the optimal cruising speed is lower than both the start and end speed; indeed this is necessary to reach the given target journey time. My comment was motivated by the presence of a braking section before coasting started, which appeared to be counterintuitive (and hence potentially worth investigating further), as eta_b < 1 means that this section represents lost energy compared to coasting only. However, I accept that your point from the response to [7.] could also apply here – i.e. that direct observations to investigate the underlying physical reasons for patterns in the results are not always possible for the algorithms. I ran some brief simulations, which suggested that the energy consumption of braking and then coasting to cruise at 13 m/s was broadly comparable to immediately coasting to 11 m/s before cruising, for a fixed distance of 18 km and journey time of around 990s. Ultimately, this is probably beyond the current scope however, and so I don’t think changes to the paper are needed here.

One other observation: it appears you have redrawn Figures 11, 13, 15 and 17, as well as changing the energy consumption and computational times for the MILP results in Table 5 compared to the first version of the paper. Do these reflect the lower values of delta_d you mention?

12. If further content is added in response to [7.], it would be worth adding brief reference to it here, and also updating the reference to Section 4.3 if the relevant discussion is moved to a larger Section 4.6.

Author Response

Dear Reviewer, 

We have included our detailed responses to your valuable comments in the attached PDF file. We appreciate your time and efforts very much which have make our paper quality much improved. 

Kind regards, 

Shaofeng 

Round 3

Reviewer 2 Report

all clear for my comments

Reviewer 3 Report

Thank you for the feedback - I fully support your changes, and am happy to recommend acceptance of the paper.